# Ecosystem responses to elevated $CO_2$ using airborne remote sensing at Mammoth Mountain, California

Kerry Cawse-Nicholson[1], Joshua B. Fisher[1], Caroline A. Famiglietti[1], Amy Braverman[1], Florian M. Schwandner[1,2], Jennifer L. Lewicki[3], Philip A. Townsend[4], David S. Schimel[1], Ryan Pavlick[1], Kathryn J. Bormann[1], Antonio Ferraz[1], Emily L. Kang[5], Pulong Ma[5], Robert R. Bogue[1], Thomas Youmans[1], David C. Pieri[1]

[1]Jet Propulsion Laboratory, California Institute of Technology, Pasadena, CA, USA
[2]Joint Institute for Regional Earth System Science and Engineering, University of California Los Angeles, Los Angeles, CA, USA
[3]United States Geological Survey, Menlo Park, CA, USA
[4]University of Wisconsin-Madison, Madison, WI, USA
[5]University of Cincinnati, Cincinnati, OH, USA

*Correspondence to*: Kerry Cawse-Nicholson, kcawseni@jpl.nasa.gov

**Abstract.** We present an exploratory study examining the use of airborne remote sensing observations to detect ecological responses to elevated $CO_2$ emissions from active volcanic systems. To evaluate these ecosystem responses, existing spectroscopic, thermal, and lidar data acquired over forest ecosystems on Mammoth Mountain volcano, California, were exploited, along with *in situ* measurements of persistent volcanic soil $CO_2$ fluxes. The elevated $CO_2$ response was used to statistically model ecosystem structure, composition and function, evaluated via data products including biomass, plant foliar traits and vegetation indices, and evapotranspiration (ET). Using regression ensemble models, we found that soil $CO_2$ flux was a significant predictor for ecological variables, including canopy greenness (Normalized Vegetation Difference Index, NDVI), canopy nitrogen, ET, and biomass. With increasing $CO_2$, we found a decrease in ET and an increase in canopy nitrogen, both consistent with theory, suggesting more water and nutrient use efficient canopies. However, we also observed a decrease in NDVI with increasing $CO_2$ (a mean NDVI of 0.27 at 200 g m$^{-2}$ day$^{-1}$ $CO_2$ reduced to a mean NDVI of 0.10 at 800 g m$^{-2}$ day$^{-1}$ $CO_2$). This is inconsistent with theory; though consistent with increased efficiency of fewer leaves. We found a decrease in aboveground biomass with increasing $CO_2$, also inconsistent with theory; but, we did also found a decrease in biomass variance, pointing to a long-term homogenization of structure with elevated $CO_2$. Additionally, the relationships between ecological variables changed with elevated $CO_2$, suggesting a shift in coupling/decoupling among ecosystem

structure, composition, and function synergies. For example, ET and biomass were significantly correlated for areas without elevated $CO_2$ flux, but decoupled with elevated $CO_2$ flux. This study demonstrates that a) volcanic systems show great potential as a means to study the properties of ecosystems and their responses to elevated $CO_2$ emissions and b) these ecosystem responses are measurable using a suite of airborne remotely sensed data.

## 1 Introduction

Terrestrial ecosystems have consistently taken up carbon over the past century, in excess or balancing losses due to deforestation and land use change, and this sink has grown with time (Le Quéré et al., 2016; Schimel et al., 2015). Much debate, however, has centred on the drivers of this uptake. Suggested mechanisms include nitrogen deposition (Peterson and Melillo, 1985), land use (Schimel, 1995), and the direct effects of carbon dioxide on plant growth (Norby et al., 2016). The last, which proposes that increased atmospheric $CO_2$ yields increased photosynthetic rates, is both the most probable and the most controversial. Although a multitude of experiments have shown positive photosynthetic responses to increased $CO_2$ consistent with the observed growth in the terrestrial sink (Drake et al., 1997), many ecologists have argued that plant growth in intact ecosystems is limited by water, light or nutrients, rather than $CO_2$ (Körner, 2006; McGuire et al., 1995).

The Free-Air Carbon Enrichment (FACE) experiments, introduced in the 1990s, allow for $CO_2$ fertilization of intact ecosystems by creating controlled fumigation conditions without the use of a growth chamber (Lewin et al., 1994). The FACE studies have been invaluable to our understanding of the $CO_2$ effect, which contributes to among the largest uncertainties in projections of Earth's climate. These studies have shown some consistent responses indicative of enhanced growth (Norby et al., 2016), as well as other physiological, morphological and ecosystem consequences, but also suffer from several structural limitations. Perhaps most notably, only short-term study periods are feasible; and it is difficult to measure slower processes like plant acclimation, shifts in species dominance induced by $CO_2$, or other long-term mechanisms mediated by changes to soil organic matter and nutrients. This is where the long-term localized emissions of volcanic $CO_2$ can play a game changing role in how to assess the long-term $CO_2$ effect on ecosystems.

As a result of limited empirical evidence for the strength of $CO_2$ fertilization effects, global carbon cycle models disagree about the significance of their associated impacts. Some models show very large $CO_2$ effects, while others indicate a smaller or saturating effect (Kolby Smith et al., 2015). Because future predicted fossil carbon uptake is highly dependent on the strength of the simulated $CO_2$ fertilization, any constraints on the long-term effect of

elevated $CO_2$ on ecosystems would be valuable in reducing uncertainty in coupled carbon-climate models (Friedlingstein et al., 2014).

Persistent diffuse volcanic $CO_2$ emissions through soils result from the degassing of magma
beneath volcanoes and offer a continuous natural experiment to study vegetation responses to elevated $CO_2$ that is expansive in both space and time. These surface discharges yield broad atmospheric enhancements that transport $CO_2$ downwind (Kerrick, 2001), resulting in swaths of variably affected plants whose periods of exposure can be over hundreds of years (Cook et al., 2001). Because volcanic $CO_2$ emissions are a vital part of the global carbon cycle (Mason
et al., 2017; Schwandner et al., 2017) and have been monitored worldwide for decades (Boudoire et al. 2017; Camarda et al., 2012; Perez et al., 2011; Gerlach, 1991), the rate and spatial distribution of these fluxes are well-understood due to an abundance of field surveys in many volcanic systems (e.g. Hernández et al., 1998; Cardellini et al., 2003; Werner and Brantley, 2003; Giammanco et al., 2007; Lewicki et al., 2014a). The "kill-zone" is the exact
location where $CO_2$ is emitted from the soil—a property of the soil being altered by the emission. Although the spatial distributions of $CO_2$ emissions within tree kill areas have been well mapped (Pickles et al., 2001; Werner and Brantley, 2003; List et al., 2005, and others), linking $CO_2$ measurements to vegetation responses along a spatially diffuse $CO_2$ degassing continuum (outside of the tree-kill zone) is a natural yet underutilized opportunity for
studying the effects of elevated $CO_2$ on plants (Schwandner et al., 2004). Furthermore, many $CO_2$ emissions in volcanic systems have been ongoing for decades or centuries, thus allowing for the observation of equilibrium, long-term ecosystem responses after transient and acclimational responses have passed.

While FACE experiments may demonstrate ecological responses to increased $CO_2$ at the outset of elevation, studies in volcanic basins can do the same on super-century scales. However, because volcanic emissions can affect entire landscapes differentially depending on the flow dynamics of the gas, they require new and innovative techniques for analysis. Remote sensing observations allow for detailed measurements across a wide spatial extent
that can be used to analyse ecological indicators of $CO_2$ effects.

Here, we present an exploratory study examining the use of airborne remote sensing data to detect ecological responses to elevated volcanic $CO_2$ emissions. It is a fundamental principal of volcanology that all active volcanoes emit $CO_2$ continuously during their entire life cycle.
We leveraged existing data over Mammoth Mountain, California – a much-studied volcano that has been passively emitting $CO_2$ at high concentrations through faults and fissures on its flanks, measured systematically since a large earthquake swarm in 1989, and their variability well documented by repeated $CO_2$ efflux mapping (Farrar et al., 1995; Lewicki et al., 2014b,

b; Werner et al., 2014). Figure 1 shows that the elevated soil $CO_2$ fluxes, measured by the USGS over a span of two decades, far exceed the atmospheric $CO_2$ measured by a flux tower at the same site.

We developed a statistical framework for examining the relationships between field measurements of soil $CO_2$ emissions into the air below the forest canopy and a suite of remotely sensed ecological variables. In this investigation, we aim to: (i) evaluate the viability of using a passively degassing volcanic system to study the properties of ecosystems; (ii) assess the detectability of ecological responses to elevated soil $CO_2$
emissions via airborne data alone; and (iii) present key lessons enabling future studies to extend our framework to other biomes. This methodology can be applied to any site that is exposed to elevated $CO_2$.

## 2 Methods

### 2.1 Data

Airborne remote sensing data from multiple sources have been acquired over Mammoth Mountain, California, USA, providing a substantial means to assess ecosystem structure (products derived from lidar, such as canopy height and biomass), composition (products derived from spectral data, such as vegetation indices and plant foliar traits), and function (data products derived from thermal data, such as evapotranspiration). Figure 2 illustrates
several of the different products used in this study, highlighting the diversity of data sources and spatial resolutions.

Mammoth Mountain is an upper montane forest ecosystem, characterised by abundant *Pinus contorta* (lodgepole pine), and also by mature stands of *Abies magnifica* (red fir), *Pinus*
*jeffreyi* (Jeffrey pine), *Pinus albicaulis* (whitebark pine), and *Juniperus occidentalis* (western juniper) (Potter, 1998). The elevation of our study areas ranged from 2700 to 2950 m. Tree-kill soils are immature High Sierra soils formed from granite, pumice, rhyolite, and obsidian parent materials (McGee and Gerlach, 1998).

### 2.1.1 Ground measurements

We investigated soil $CO_2$ fluxes within five actively degassing areas on Mammoth Mountain documented by Werner et al. (2014) in 2011 and 2012, which represents a period of relatively high emissions (up to 2000 g m$^{-2}$ day$^{-1}$ of $CO_2$). As described by Werner et al. (2014), fluxes were measured along fixed grid points using the accumulation chamber method (Rahn et al.,

1996). In situ measurements were obtained using a West Systems® (Florence, Italy) portable fluxmeter equipped with a LI-COR820 infrared gas analyser. Based on statistical analysis, Werner et al. (2014) found soil $CO_2$ fluxes measured within areas of volcanic $CO_2$ emissions to be significantly elevated over background areas that were dominated by soil respiration of

$CO_2$. Maps of soil $CO_2$ flux were simulated from in-situ measurements at 1 m resolution using a sequential Gaussian simulation algorithm by these authors and we resampled their data to the Airborne Visible/Infrared Imaging Spectrometer (AVIRIS) resolution (13 m) using nearest neighbour resampling. Conventionally, studies of diffuse soil degassing of $CO_2$ on volcanoes have emphasized understanding of the modes, locations, geometries, and

changes in volcanic flank degassing for purposes of vulcanological research, hazard assessment, and monitoring. In many cases, volcanologists have focused on areas associated with sufficient emissions of heat and $CO_2$ that vegetation has been killed off. In this study however, we focused on vegetated areas where somewhat more mildly enhanced levels of volcanic $CO_2$ emissions into the forest ecosystems might be beneficial for plant growth,

rather than adverse.  As such, we investigated zones and gradients *around* tree-kill areas, excluding areas that were barren or contained dead trees by filtering by fractional vegetation cover, where appropriate.  The tree-kill areas have local soil conditions that are not representative of the larger ecosystem. In addition, because tree-kill areas on Mammoth Mountain are largely associated with "cold" $CO_2$ emissions, we completely avoided

confounding influences of hydrothermal heat or acidic vapour emission on ecosystem response. Indeed, there is no significant $H_2S$ nor any $SO_2$ present at soil levels at this site, and $CO_2$ makes up ~99% of the gas by volume; see, for example, data in (Sorey et al 1998, Werner et al, 2014), and a number of papers on volcanic degassing at Mammoth Mountain (Lewicki 2006, 2007, 2008, 2012, 2014). The remaining 1% is made up of $N_2$ and $O_2$.

The use of a high-spatial-resolution time-averaged (to limit the influence of varying meteorological conditions) map of canopy-level atmospheric $CO_2$ concentration would be most applicable to assess ecosystem response to elevated atmospheric $CO_2$ concentrations. However, such maps are unavailable. We therefore took advantage of the extensive record of

soil $CO_2$ fluxes available for Mammoth Mountain. Although the effects of elevated $CO_2$ in the soil may be difficult to de-convolve from elevated $CO_2$ in the atmosphere, we treat their effects uniformly.  Implications of this are discussed below.

Although the airborne datasets cover a wider region, only points with associated soil $CO_2$ flux measurements were used to derive our models. The $CO_2$ flux measurements were spatially resampled to match the resolution of the other datasets, which resulted in small estimations with low confidence along the edges. To avoid spurious model fits, edge points

with $CO_2 < 5$ g m$^{-2}$ d$^{-1}$ were excluded, where the $CO_2$ range is [0,2000] g m$^{-2}$ d$^{-1}$. In the remainder of this manuscript, analysed points with elevated $CO_2$ flux will be referred to as $eCO_2$.

### 2.1.2 AVIRIS

The Airborne Visible/Infrared Imaging Spectrometer (AVIRIS) Classic instrument acquires data from 400 to 2500 nm in 224 contiguous spectral channels. AVIRIS imagery was acquired over Mammoth in October 2014; this flight was chosen from a number of possible surveys of the area to minimize snow cover, and also because of its temporal proximity to the $eCO_2$ ground measurements. The standard level 2 (L2) atmospherically corrected reflectance
data (Thompson et al., 2015) was used (available from https://aviris.jpl.nasa.gov/), and the data had a spatial resolution of 13 m. This data was collected as part of the NASA HyspIRI Preparatory Airborne Campaign.

*Vegetation indices*
Vegetation indices are commonly used as an indicator of vegetation health and/or greenness. While many vegetation indices are related, they differ enough to be considered independent variables. E.g. some account for soil moisture, others weight plant greenness more heavily. This was an exploratory effort in investigating the effects of $CO_2$ on any measure of plant function, composition, and structure, and so we attempted to cover all avenues of
investigation. The following indices were derived from the AVIRIS spectral data:
- The Normalized Difference Vegetation Index (NDVI)
- Simple Ratio Index
- Enhanced Vegetation Index
- Red Edge Normalized Difference Vegetation Index
- Modified Red Edge Simple Ratio Index
- Modified Red Edge Normalized Difference Vegetation Index
- Vogelmann Red Edge Index 1

Each uses a ratio between narrow bands to represent vegetation health as a single index, and all are described more fully in (Thenkabail et al., 2016).

*Foliar traits*
The chemical composition of plants affects light interactions, especially in the short-wave infrared (Singh et al., 2015). Therefore, imaging spectroscopy can be used to map key vegetation properties, especially those affecting carbon and nutrient interactions. Spectral
features, derived from data such as AVIRIS, have been shown to correlate significantly with certain chemicals and plant properties, such as carbon, nitrogen, nitrogen isotope 15, Leaf

Mass per Area (LMA), cellulose, and acid digestible lignin (Singh et al., 2015). These properties are associated with photosynthesis, light-harvesting ability, nutrient fluxes, and can be used to characterise vegetation responses to disturbances or climate trends (Townsend et al., 2008).

The data were first corrected for its bi-directional reflectance distribution function (BRDF), using the Ross-Thick BRDF model with a quadratic volumetric scattering term (Roujean et al., 1992; Lucht et al., 2000). In situ vegetation chemical measurements, along with propagated uncertainties, were used to derive partial least squares regression models for each
trait. Since these equations were derived in the nearby area of the Sierra Nevada Mountains, these equations were applied to the BRDF-corrected AVIRIS data used in this study.

Infeasible negative numbers were removed for the modelling.

### 2.1.3 MASTER

The MODIS/ASTER (MASTER) airborne simulator acquires data in 50 channels between 0.4 – 13 μm. We utilized the five thermal channels (10 – 13 μm), which had been processed to Level 2 (available from https://master.jpl.nasa.gov/). MASTER data were acquired in November 2013, with a 50 m spatial resolution.

*Land Surface Temperature*
The five thermal bands from MASTER were used to calculate Land Surface Temperature (LST) in a standard Level 2 product. The acquired data were processed to radiance using MODTRAN 5.2 for the atmospheric correction, along with a water vapour scaling method (Tonooka, 2005). The Temperature Emissivity Separation (TES) algorithm was then used to
derive LST and spectral emissivity (Gillespie et al., 1998).

The MASTER data are at coarser spatial resolution (50 m) compared to the other datasets (e.g., the working resolution for reprojection is the AVIRIS resolution of 13 m). An ideal dataset would have MASTER acquired at 13 m, or similar (~10 m; i.e., the scale of an
individual tree canopy), but in order to build a comparable dataset for this analysis, we used two resampling methods: the standard nearest neighbour resampling; and a statistically principled method proposed in Ma et al. (2018). The statistical model proposed by Ma et al. (2018) represented LST as a combination of low-dimensional random effects linked with basis functions and a Gaussian graphical model (also called Gaussian Markov random field).
As demonstrated by the Ma et al. (2018), this model provides a flexible and computationally efficient way to characterize potentially complex and nonstationary spatial variability. The parameters of the underlying statistical model were fitted to MASTER LST and ET data at 50

m resolution, using maximum likelihood estimation via an Expectation-Maximization (EM) algorithm. The resampled data at 13 m spatial resolution were then generated via conditional statistical simulation in which we required that when aggregated back to the original coarse resolution, the resampled data matched the original MASTER data exactly.

*Evapotranspiration*

Evapotranspiration (ET) is the key water variable in ecosystem functioning, indicating plant water use and loss (Fisher et al., 2017). In this study, ET was calculated using the PT-JPL retrieval (Fisher et al., 2008), which partitions ET into canopy transpiration, soil evaporation, and interception evaporation by transforming potential ET (Priestley and Taylor, 1972) into actual ET using ecophysiological constraints. The ECOSTRESS ET retrieval system was used to incorporate MASTER LST as the thermal input (Fisher et al., 2015); additional ancillary data were incorporated from MODIS and Landsat to constrain meteorological and phenological controls on ET (Verma et al., 2016; Famiglietti et al., 2018; Ryu et al., 2011; Kobayashi et al., 2008). The final ET product used here was only the canopy transpiration component (referred to as ET throughout), as our analytical interest lies only in the vegetation response to $eCO_2$.

## 2.1.4 ASO

The Airborne Snow Observatory (ASO, http://aso.jpl.nasa.gov) is a coupled lidar (Riegl Q1560) and spectrometer (CASI-1500) mounted on a King Air A90 aircraft, and was originally developed to monitor snow in the mountains for water resource management (Painter et al., 2016). The Riegl Q1560 is a dual scanning lidar with two 1064 nm laser sources; each scanner is tilted in the along-track direction by $\pm 8°$ and the cross-track direction by $\pm 14°$ for enhanced retrieval of vertical surfaces. On June 27, 2017 ASO surveyed Mammoth Mountain, retrieving comprehensive lidar point cloud data at a mean of 7.8 pt. $m^{-2}$ (max. value ~60 pt $m^{-2}$). Riegl RiPROCESS software was then used to a) extract point cloud data from raw waveforms (RiANALYZE) using the RiMTA Multiple Time Around algorithm and the RLMS Simple Classification Procedure for classification (SCP1), b) georeference the point cloud (RiWORLD), and c) export the point cloud to LAS 1.2 in UTM projection (RiWORLD).

*Digital Terrain Model*

The ASO lidar point cloud data were filtered to remove outliers by applying an elevation filter to eliminate points that exceed $\pm 100$ m from a baseline digital terrain model (DTM) that was obtained from the USGS (United States Geological Survey). The ASO data processing chain includes the identification of ground and off-ground points using the Multiscale Curvature Classification algorithm (Evans and Hudak, 2007) and the calculation of a DTM (3

m x 3 m) that corresponds to the bare soil surface as interpolated from the lidar points classified as ground. Any data voids were then in-filled using search windows that were centred on each void pixel.

*Slope and Aspect*
The slope (steepness) and aspect (direction) were derived directly from the DTM with the terrain analysis processing tool provided by QGIS. These geo-algorithms use a first-order derivative estimation to calculate the slope angle for each pixel in degrees relative to the horizontal plane and the slope exposition in degrees counter-clockwise from north.
Aspect was processed to account for circular angles, by considering:
$$K1 = \ \sin(\alpha + (90 - d)) + 1 \quad\quad\quad\quad\quad (1)$$
$$K2 = \ \cos(d - \alpha) + 1 \quad\quad\quad\quad\quad (2)$$
where $\alpha$ is the aspect derived from the DTM as described above, and $d$ is the prevailing wind direction. In the absence of local data, we assumed the prevailing wind direction to be 270°
(e.g. Anderson and Farrar, 2001; Lewicki et al., 2008; Lewicki and Hilley, 2014). (Note, the results presented below were not sensitive to this assumption.)

*Canopy Height and Biomass*
The aboveground biomass (AGB) map (30 m x 30 m) was calculated by integrating ASO
lidar measurements on forest structure and field inventory data into an allometric equation developed by Garcia et al., (2017):
$$AGB = 11.50 \times MCH^{1.20} \times FC^{0.88} \quad\quad\quad\quad\quad (3)$$
where MCH and FC are lidar-derived maps of mean canopy height and fractional cover, respectively. Eq. 3 was calibrated using AGB reference values derived from 69 field
inventory plots located in the Stanislaus National Forest and Yosemite National Park, Sierra Nevada, California. To compute the lidar-derived maps, we first normalized the ASO lidar point cloud to calculate the effective height of vegetation by removing the effect of topography using the DTM described here above. Then, we used the normalized point cloud to calculate a canopy height model (CHM, 1m x 1m) by selecting the highest lidar point
within each grid cell. Finally, the MCH was calculated by averaging the CHM within each 30 m cell, whereas the FCC was computed as the ratio of grid cells covered by vegetation (i.e. MCH>2 m) to the total number of cells. Note that we defined both MCH and FC with a grid cell size of 30 m in order to agree with the size of the field samples (Garcia et al., 2017). We assumed that Eq. 3 was transferable to our study site because the calibration plots are located
only 80 km apart and they are both populated by vegetation of the upper montane and subalpine biotic zones.

### 2.1.5 Compiling the Dataset

The data were first processed to create derived products, and then geolocated to the AVIRIS native resolution of 13 m. That is, for each AVIRIS pixel, the other datasets were resampled and reprojected so that every pixel is associated with a vector of remotely sensed and derived
values. Datasets with finer resolution (soil $CO_2$ flux and lidar) were averaged using the nearest neighbour principle. Derived products with coarser resolution (fractional cover, biomass, and evapotranspiration) were resampled using nearest neighbour resampling (e.g. the same biomass value may cover multiple AVIRIS pixels). Because its pixels were the largest, ET was also resampled using a statistically based method, described above in Section
2.1.3. We note that although the downscaling approach is robust and statistically sound, we acknowledge that our statistical estimates involving ET will include some uncertainty due to spatial resolution.

Once all pixels had been resampled, we had a total of 5520 data points. For certain
experiments we found it necessary to threshold by fractional vegetation cover (FC>0.7; n=55), although the full dataset was used wherever possible.

The dates of acquisition also differed across datasets. The soil $CO_2$ flux datasets used in this study were measured during a peak in $CO_2$ emissions (Werner et al., 2014), and this peak in
emissions is thought to affect future plant growth. However, we are observing a snap shot of vegetation function within a zone small enough to be influenced by the same meteorological inputs, and our models have accounted for confounding factors such as slope, elevation, and aspect. Therefore, we considered measurements to be relative on a spatial scale, by comparing neighbouring pixels. The topographic confounders and the fractional cover are
derived from the lidar data acquired four years after the MASTER data; however, we do not expect changes in the terrain during that time period, and tree presence is unlikely to have changed significantly.

### 2.2 Statistical Modelling

The variables assessed included: vegetation indices; plant foliar traits; evapotranspiration;
canopy height; and, biomass. Given this combination of variables, we tested whether changes in eCO2 were associated with significant changes in vegetation. We performed a series of multiple linear regressions using eCO2 as a predictor of various vegetation variables; in particular, regression ensembles build collections of linear regression models, utilizing different predictor combinations, including multiplication of predictor variables. To control
for confounding variables including elevation, slope, and aspect (which are topographic

proxies for temperature, moisture, and light, respectively), we included them as predictors in the model. Then, the regression coefficient estimate for $eCO_2$ is an estimate of the change in the response variable due to a change in $eCO_2$, holding all other variables in the model (the confounders) constant. Random forests were investigated, and found to produce similar results. For ease of interpretation, we present here the results of the linear regression ensembles.

Fractional vegetation cover (FC; derived from the lidar) was considered a proxy for vegetation presence. The geometric variables elevation, slope, and aspect were also derived from the lidar point cloud, as described above. Figure 3 illustrates the stratified behaviour of NDVI as coloured by the four confounding variables. There is a particularly clear separation for fractional cover, which reinforces an expected result: $eCO_2$ had negligible effect on vegetation indices and other variables over bare ground, but showed higher impacts on fully vegetated pixels. Therefore, we model each vegetation variable, $V$, as

$$V = b_1 C + b_2 F + b_3 S + b_4 A + b_5 E + f(C, F, S, A, E) + \varepsilon \qquad (4)$$

where $C$ is the elevated soil $CO_2$ flux, $F$ is the fractional vegetation cover, $S$ is the slope, $A$ is the aspect, $E$ is the elevation, and $\varepsilon \sim N(0, \sigma^2)$ is random error. The function $f(\cdot)$ describes relationships between the predictor variables, which for this model is limited to the first order interactions:

$$f(C, F, S, A, E) = b_6 \, C \cdot F + b_7 \, C \cdot S + b_8 \, C \cdot A + b_9 \, C \cdot E + b_{10} \, F \cdot S + \cdots \qquad (5)$$

Our hypothesis is $H_A: b_1 \neq 0$, that is, that the effect of $eCO_2$ on vegetation variable $V$ is different from zero. Our null hypothesis is then $H_0: b_1 = 0$.

Certain other confounding variables may affect the modelled relationships. The following scenarios and/or variables were also tested as confounders, but did not affect the model outcome: pixel position; site number; and species (plant species were estimated by performing an unsupervised classification on the AVIRIS data). The $eCO_2$ dataset was also shifted to simulate winds and atmospheric pressure (Ogretim et al., 2013). This did not have an impact on the results.

Additionally, diurnal patterns of mountain slope air flows may dilute and enrich the bulk air mass the trees are exposed to with respect to $CO_2$ concentrations (Pypker et al., 2007). If these air flow patterns are strong, they may drain the local $CO_2$ enhancement during morning and evening hours, when these flow events are usually strongest. However, due to the

constant nature of these localized enhanced emissions, the gradient, if it was diluted by such effects, re-establishes itself during calmer daytime and night-time hours, as is evident by the volcanic diffuse $CO_2$ emission signal being detectable from airborne in-situ measurements above the investigated sites as well (Gerlach et al. 1999).

When evaluating the dynamics between different variables, it is assumed that the study from which our ground measurements were derived (Werner et al., 2014), covered most of the known $CO_2$ diffuse emission areas, and so the remainder of the scene exists as a control. The control pixels were also thresholded according to the range of the confounding variables found for the $eCO_2$ points. Therefore, we considered only control points with elevation, slope, and aspect values, respectively, between 2700 and 2950 m, less than 30°, and less than 350°.

## 3 Statistical Estimation

Although the models were run for 42 explanatory variables (including additional vegetation indices, foliar traits, and other vegetation descriptors), for the sake of brevity we only present the best performing variables (traits with significant p-values are shown, and for all other variables, those with significant p-values and $R^2 > 0.5$). For the variables shown in Table 2, the p-value of the $eCO_2$ term, $b_1$, was for each model $< 0.05$, and in most cases $\ll 0.05$. The most significant predictor was determined by ordering terms by p-values.

As the confounding variables are expected to drive the behaviour of ecosystem properties, a reduced $eCO_2$ "rank" (in terms of p-value significance) does not negate the impact of $eCO_2$ in the models; in fact, each ecosystem variable was strongly influenced by increasing $eCO_2$, given the low p-values for the $eCO_2$ coefficient in each model. The rank of each predictor variable is given in Table 3. Since multiplicative terms are allowed, two terms in a single ranking column (say, slope and fractional cover) means that the multiplication between the two terms is the term with the lowest p-value. To reduce the complexity of the table, each variable is listed only once, in order of first appearance, whether singly or as a product.

## 4 Results

### 4.1 Structure: Canopy Height and Biomass

Canopy height and biomass were accurately modelled with high $R^2$, as seen in Table 2 and Figure 4, although $eCO_2$ was the least significant predictor. In each case, $eCO_2$ was still

regarded as statistically significant, but had lower predictive power than the topographic variables.

Figure 5 shows the predictor variable $eCO_2$ against the predicted biomass. There is variability
at low $eCO_2$ levels, but overall a small decrease in biomass with increasing $eCO_2$. This decrease appears to saturate, and is better fit by a logarithmic function, however, given that interactions between terms in the model is allowed, we do not necessarily expect a linear fit, since the $eCO_2$ contribution to the model may be multiplied by other confounding variables. There is also a decrease in biomass variance. In other words, trees exposed to higher $eCO_2$
are more similar.

### 4.2 Composition: Vegetation Indices and Foliar traits

The performance of different vegetation indices and foliar traits varied. NDVI was best modelled ($R^2 = 0.68$), and with $eCO_2$ as the most significant predictor (p-value of 1e-12). In general, the indices were better modelled than the traits.
Figure 6 shows the predicted model for NDVI (a) and the canopy nitrogen concentration trait (b) against the $eCO_2$ predictor variable. Modelled NDVI decreases with increasing $eCO_2$, and there is a decrease in variance with increasing $eCO_2$. The modelled canopy nitrogen concentration trait increases with increasing $eCO_2$.

### 4.3 Function: Evapotranspiration

Canopy transpiration was relatively well represented by the $eCO_2$ model with an $R^2 = 0.55$. For comparison, total ET was not well represented by the $eCO_2$ model ($R^2 = 0.23$), which is sensible, as $eCO_2$ is expected to affect only plant transpiration and not soil evaporation. $eCO_2$ was the second most significant predictor, with fractional vegetation cover the most significant. Given that MASTER data were originally acquired at a much coarser resolution
(50 m) than the $eCO_2$ ground data (1 m), and that both were resampled to 13 m resolution for the overall consistent analysis, there may have been error introduced due to the resampling. This effect is seen by the much lower model fit with the statistical resampling, although the predicted models follow the same trend. In the remainder of the manuscript, references to ET refer to the data resampled using nearest neighbour resampling.
Figure 7 shows the ET predicted by the model for the predictor variable $eCO_2$. There is a decrease in ET for increasing $eCO_2$, along with a decrease in variance.

### 4.4 Ecosystem synergies

Given that many of the vegetation indices and traits are only appropriate in the presence of vegetation, a fractional cover threshold of 0.7 was used for the $eCO_2$ sample, for the sake of evaluating the dynamics between modelled variables. With this threshold, only 55 data points remained, and so the sample size is too small to make claims of statistical significance. Therefore, we present the following results as interesting observations that may inform future data acquisition.

Figure 8 shows the dynamics between variables in the entire scene (i.e., non-elevated, background soil $CO_2$) versus the points with $eCO_2$ measurements. It is important to note that in each sub-figure, independent data sources are used to avoid showing intrinsically correlated datasets. Fractional cover and biomass are derived from the ASO lidar data; the vegetation trait data and foliar traits are derived from AVIRIS imagery; and ET is derived from MASTER data. In this case, the variables shown are directly as observed (or derived directly from the data source).

We observed interesting dynamics between ecosystem variables, suggesting great potential for future research. In the $eCO_2$ subset, NDVI was, on average, lower than that observed for the same fractional cover in the control dataset (Figure 8 a). This is consistent with the model illustration of decreased greenness for increasing $eCO_2$. Similarly, ET was lower in the $eCO_2$ subset for pixels with the same NDVI observed in the control, showing a greater degree of stress even when plants have the same greenness (Figure 8 b). In addition, the strong linear relationship between ET and NDVI appears to break down for the points affected by $eCO_2$.

Canopy nitrogen in the $eCO_2$ subset increased with fractional cover, unlike the control which remained flat, which again mimics the modelled data findings (Figure 8 c). ET was lower in the $eCO_2$ subset for the same biomass, which implies that plants are doubly affected by the enhanced $CO_2$ – the biomass decreases with increasing $eCO_2$, and the ET decreases further with decreasing biomass (Figure 8 d). Again, the strong linear relationship between ET and biomass breaks down for those points affected by $eCO_2$. These findings suggest complex relationships between ecosystem parameters in their response to increasing $eCO_2$.

### 5. Discussion

Using airborne remotely sensed ecosystem properties against a ground measured database of $eCO_2$ (volcanic excess $CO_2$ emanating into the forest canopy through the soil), we evaluated the effects of increasing $eCO_2$ on plant structure, function, and composition. Our aims were

to: (i) evaluate whether a passively degassing volcanic system is a viable means to study properties of ecosystems; (ii) determine if ecosystem variables are adequately detected using airborne data; and (iii) present key lessons learnt that can enable similar studies over different biomes.

This study has provided initial observations of ecological responses to $eCO_2$ that are measurable from airborne data. We found that: a) $eCO_2$ was a significant predictor in regression ensemble models of ecosystem variables, and b) there were visual differences between the sites of increased $eCO_2$ and the background image. This work also demonstrates

that an active volcanic system is a viable way in which to study the $CO_2$ effect on ecosystems.

The regression ensemble model showed that $eCO_2$ was a significant predictor for two structural variables (canopy height and biomass), nine composition variables (6 vegetation

indices, 3 foliar traits), and a function variable (ET). Therefore, as hypothesized, $eCO_2$ affects ecosystems in structure, composition, and function, all of which are detectable both with airborne observations as well as within a volcanically-derived $eCO_2$ system. Further evaluation of the model showed that both canopy height and biomass decreased with increasing $eCO_2$; the vegetation indices decrease with increasing $eCO_2$; canopy nitrogen

concentration increases; LMA decreases; Carbon decreases; and ET decreases.

Some of these observations contrasted with results found in other published studies, while others agreed. For instance, our study found a decrease in NDVI with increasing $eCO_2$, which correlates to the multispectral satellite findings of Rouse et al. (2010) and Cholathat et al.,

(2011). In some cases, the decrease others have found can be explained by the tree-kill effect, where vegetation is removed. However, by accounting for fractional cover in our models, we have shown that NDVI decreases independently from fractional cover (see in Figure 8). This shows that, regardless of whether the number of trees changes, the greenness of individual trees is reduced. This finding is in direct contrast with the $CO_2$ fertilization hypothesis which

states that rising $CO_2$ has a positive effect on plant growth and productivity due to increased availability of carbon, and which has been shown using field data (Huang et al., 2007; Zhu et al., 2016). However, this decrease in NDVI could also be explained by a reduction in leaves, rather than a reduction in leaf health, due to more efficient leaves (e.g., higher nutrient concentration, more efficient in water use).

The decrease in canopy height and biomass agrees with the tree-ring study done by Biondi and Fessenden (1999), which also found slower Lodgepole Pine growth rates in high $CO_2$ emission areas on Mammoth Mountain. However, a study by Smith et al. (2013) found an

increase in biomass in the mixed-species temperate forest FACE experiment. In that experiment, there was large variation between and within species, and the experiment was limited to four years. Perhaps a long-term species composition shift due to $eCO_2$ was the cause of the change in biomass in our study, but we do not have individual tree species-level
data to support this hypothesis.

Our model showed an increase in canopy nitrogen, which could indicate species selection or individual plant optimization, given the decrease in NDVI, biomass, and ET. Canopy nitrogen is associated with plant's investment in photosynthesis (Singh et al., 2015). We also
found an increase in canopy nitrogen relative to fractional cover, showing that the change in nitrogen was not impacted by an increase in overall vegetation for those sites (Figure 8). Tercek et al. (2008) noted that *Dichanthelium lanuginosum* (hot springs panic grass) in Yellowstone had made physiological adjustments to photosynthetic enzymes in response to long-term exposure to $CO_2$, and a study of ice cores showed a 40% decrease in stomatal
density over the last 200 years, which paralleled an increase in global $CO_2$ (Woodward, 1987).  However, Sharma and Williams (2009) evaluated vegetation naturally exposed to $CO_2$ in Yellowstone National Park, and found reduced nitrogen at a leaf level in *Pinus contortus* (Lodgepole Pine), and increased nitrogen at a leaf level for *Linaria dalmatica* (Dalmation Toadflax; an invasive, non-native herb). Once again, the species-level differences
highlight the need for remote sensing analysis over areas that encompass wide species variation, in order to understand overall trends.

Kimball et al. (1998) found a slight increase in ET in a FACE experiment over cotton fields, but that increase was within the error of the ET estimation, and so was not deemed
statistically significant. In contrast, Nendel et al. (2009) found a decrease in ET, and an increase in dry above-ground biomass over a FACE crop rotation experiment. In this study, we found a decrease in ET. In addition, we found a decrease in ET relative to both NDVI and biomass, when comparing the points affected by $eCO_2$ to those unaffected points in the surrounding area. The unaffected sites showed a positive linear relationship between ET and
both NDVI and biomass, which appeared to break down for points affected by $eCO_2$ in both relationships.

The combination of lower NDVI, higher canopy nitrogen, and higher ET suggests a canopy that uses less water with rising $CO_2$  resulting in higher water use efficiency, with a nutrient
rich canopy. Since leaves are stronger and more efficient, fewer are required for photosynthesis. While biomass increased slightly, the more obvious change was the decrease in the variance of biomass, which points to alignment to more similar trees with elevated $CO_2$.

High fluxes of $CO_2$ through soils in "kill zones" on Mammoth Mountain have likely impacted forest ecosystems through oxygen deprivation in soil pore space, inhibition of root respiration and soil acidification (Farrar et al., 1995; Qi et al., 1994; McGee and Gerlach, 1998). Since

we used soil $CO_2$ flux as the predictor variable in the model, some of the observed ecosystem responses may therefore be due to the effects of high concentrations of $CO_2$ on the soil environment or some combination of soil and atmospheric effects. However, by using fractional cover as an input to the model, and excluding the "kill zones" altogether to derive Figure 8, we are focusing on the $CO_2$ gradient over vegetated areas around these zones, that

are unlikely to be affected by soil acidification. The Mammoth Mountain soil $CO_2$ flux dataset does, however, provide a record of $CO_2$ emissions that is more stable in space and time than measurements of atmospheric $CO_2$ concentrations. In particular, forest canopies will through time be exposed to $eCO_2$ at highly variable levels, because the originally mostly invariant $eCO_2$ once emitted through the soil into the sub-canopy atmosphere, is subject to

highly variable dispersion from thermal and wind disturbances at minute, diurnal, and seasonal scales (Staebler and Fitzjarrald, 2004). In-canopy concentration measurements of $eCO_2$ will therefore be highly variable, and especially if conducted instantaneously, may not be representative of the long-term relative exposure strength in the canopy.

Vegetation at this site is also responding to the partial pressure of $CO_2$ in the atmosphere, among other gases. A response above the asymptote of the net photosynthetic rate versus internal $CO_2$ partial pressure (A-Ci curve) would result in very little vegetation response to the partial pressures (Tissue, Griffin, and Ball 1999). However, the partial pressure even at elevated molar concentration at Mammoth are about 60% of those at sea level. The fact that

we see systematic ecosystem effects suggests that elevation is not on the flat part of the A-Ci curve. In other words, even if elevation were to reduce the $CO_2$ effect, we still are seeing strong $CO_2$ effects regardless, highlighting just how important and strong of a response we are able to detect.

We will clarify that the effects should not necessarily be given a subjective description of 'negative'; rather, it is important to note that the $CO_2$ fertilization effect is unlikely to continue indefinitely, particularly at the same rates that FACE studies have shown only in the short-term. All other experiments have been unable to show long-term effects. Our study suggests that over the scale of decades, some of these hypothesized greening or biomass

increases may not be sustainable. Other results, such as an increase in canopy nitrogen with increasing $CO_2$, do seem to remain consistent with our study, however.

This exploratory study leveraged existing data acquired over Mammoth Mountain. We used ASO lidar, AVIRIS, and MASTER data to derive products that describe ecosystem structure, composition, and function, and used field $eCO_2$ measurements to show that elevated $CO_2$ was a significant predictor of ecosystem variables, including vegetation indices, plant foliar traits,
biomass, and evapotranspiration. While our study has shown the promise of airborne remote sensing in detecting measurable ecosystem changes in forest ecosystems on and around a $CO_2$-emitting volcanic system, it was also completed using an existing ad-hoc collection of data. The nature of the collection of data sources enabled us to understand the details of the data characteristics necessary for future studies.

While this study is useful for showing the benefit of both a passively emitting volcanic system and airborne data for evaluating the ecosystem response to $eCO_2$, we anticipate that more meaningful results would be obtained with all datasets acquired simultaneously, at the same resolution. ET in particular varies over short time periods due to the influence of
meteorological inputs, and so multi-temporal acquisitions would provide a better overview of the ecosystem function. Other possible factors could affect this very complex system, some of which have been discussed previously, including pH, oxidative stress, water/nutrient availability, extreme climate events, and plant vigour. More fieldwork and test sites would be needed to rule out many of these factors. Other data, such as photosynthesis, may also add to
future analysis. We note that this study was exploratory, and that this study was intended to identify both potential signals as well as design elements for further study.

## 6. Conclusions

This exploratory study used airborne remote sensing data, coupled with ground measurements of soil $CO_2$ flux on a forested volcano, to derive relationships between rising
$CO_2$ emissions and ecosystem structure, function, and composition metrics. We have shown that passively emitting volcanic systems are viable environments in which to study $CO_2$ impacts on ecosystems, with $eCO_2$ the most significant predictor in regression ensemble models of several ecological variables, including NDVI, canopy nitrogen concentration, ET, and biomass. When comparing differences between vegetation parameters affected by $eCO_2$
and those estimated over the background scene, we found contrasting patterns and dynamics between ecological variables, showing that a combination of different remote sensing platforms is capable of providing a comprehensive view of ecosystem responses to long-term elevated volcanic $CO_2$.

Key lessons learnt from this study include:

1. Future campaigns should acquire all data at the same or similar resolution, at individual tree-scale
2. Well more than 55 vegetated tree points are necessary in order to draw meaningful conclusions regarding the dynamics between variables in Mammoth Mountain (which has one dominant tree species). The number of required points in other environments will vary according to ecosystem complexity, and will likely far exceed this number.
3. Combining lidar and spectral data across a range of wavelengths yielded a more complete view than using any one data source alone.

## 6. Acknowledgements

We thank Cynthia Werner for providing soil $CO_2$ flux data over our study area; Gregory Halverson for processing ET data; Zhiwei Ye for processing functional trait data; and Anthony Bloom, Stuart Wilkinson, Christoph Kern, and Deborah Bergfeld for discussions during earlier stages of this project. The research described in this paper was carried out at the Jet Propulsion Laboratory, California Institute of Technology, under contract with the National Aeronautics and Space Administration. © 2018 California Institute of Technology. Government sponsorship acknowledged.

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

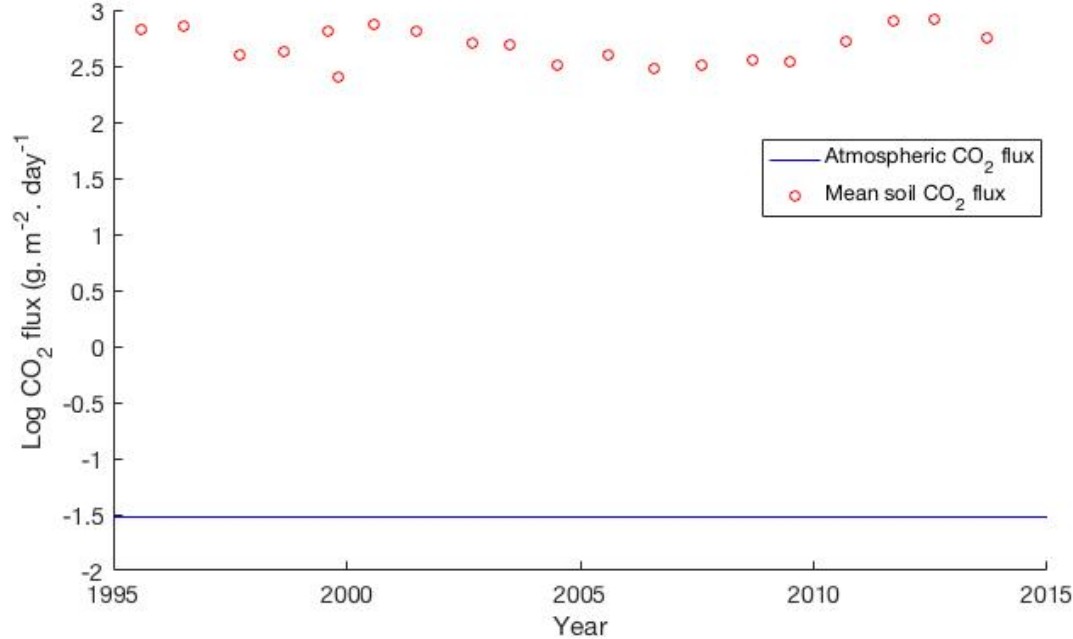

**Figure 1: The USGS has measured elevated soil CO₂ flux at Horseshoe Lake for the past two decades (Werner et al., 2014). These values are consistently higher than the atmospheric CO₂ measured by USGS California Volcano Observatory eddy covariance**
5   **station at Horseshoe Lake at the time of AVIRIS overpass on October 21, 2014 (indicated by a solid line for clarity within the figure).**

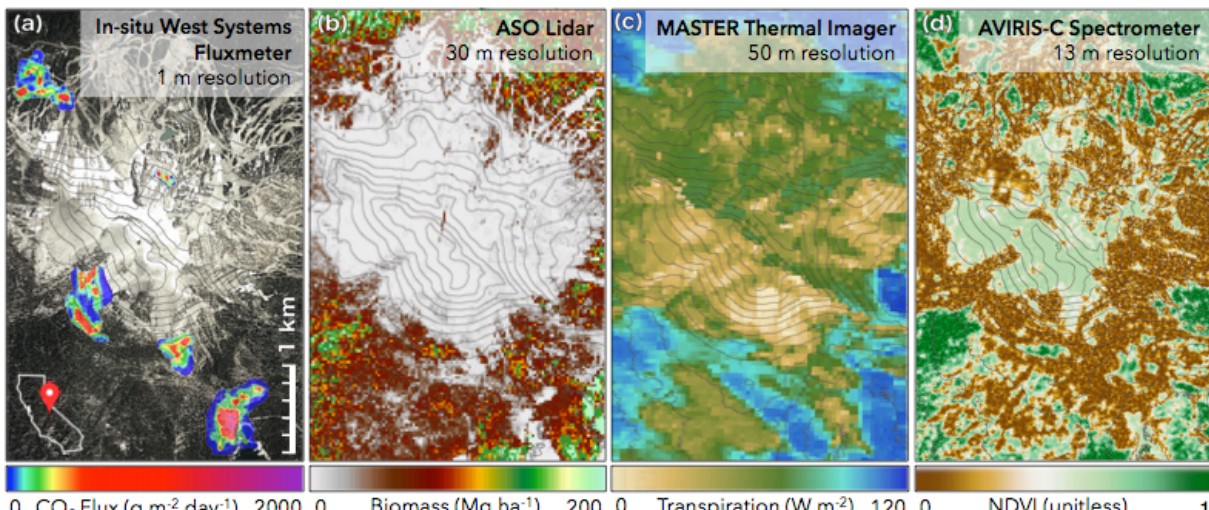

**Figure 2: A wealth of remotely sensed imagery has been acquired over Mammoth Mountain. Some data products used in this study include (a) maps of soil $CO_2$ flux simulated based on accumulation chamber measurements, shown overlain on aerial RGB image; (b) above-ground biomass derived from Airborne Snow Observatory (ASO) lidar; (c) evapotranspiration derived from the MODIS/ASTER (MASTER) airborne simulator; and (d) Normalized Difference Vegetation Index (NDVI) derived from Airborne Visible/Infrared Imaging Spectrometer (AVIRIS image).**

**Table 1: Data sources are shown along with the year in which they were acquired, the original resolution of the dataset, and the method by which it was resampled. All datasets were resampled to the AVIRIS resolution of 13m.**

| Data source | Year acquired | Original resolution | Resampling method |
| --- | --- | --- | --- |
| Soil $CO_2$ flux | 2011-2012 | 1 m | Nearest neighbour |
| Canopy height | 2017 | 1 m | Nearest neighbour |
| Vegetation indices | 2014 | 13 m | Original resolution |
| Foliar traits | 2014 | 13 m | Original resolution |
| Fractional cover | 2017 | 30 m | Nearest neighbour |
| Biomass | 2017 | 30 m | Nearest neighbour |
| Evapotranspiration | 2013 | 50 m | Nearest neighbour; Ma et al. (2018) resampling |

**Table 2: The best performing vegetation indices (VI) and traits are shown with the predictive significance of eCO$_2$ in the model, and with their correlation with a regression ensemble that included elevation, slope, aspect, and fractional cover as confounding variables (n=5520). The most significant predictor was determined by ordering terms by p-values.**

| Variable | Most significant predictor term | Estimate for eCO$_2$ coefficient | Standard error for eCO$_2$ coefficient | p-value for the eCO$_2$ term | Model R$^2$ |
|---|---|---|---|---|---|
| *Structure:* | | | | | |
| Canopy height | Slope, FC | 6e-3 | 1e-3 | 4e-6 | 0.92 |
| Biomass | FC | 1e-1 | 2e-2 | 5e-6 | 0.83 |
| | | | | | |
| *Composition (Vegetation indices):* | | | | | |
| Normalized Difference Vegetation Index (NDVI) | eCO$_2$ | -6e-5 | 8e-6 | 1e-12 | 0.68 |
| Red Edge Normalized Difference VI | eCO$_2$ | -3e-5 | 6e-6 | 1e-9 | 0.67 |
| Modified Red Edge Normalized Difference VI | eCO$_2$ | -7e-5 | 6e-6 | 2e-27 | 0.65 |
| Vogelmann Red Edge Index 1 | eCO$_2$ | -3e-5 | 1e-5 | 2e-3 | 0.64 |
| Enhanced Vegetation Index | eCO$_2$ | -1e-4 | 1e-5 | 2e-22 | 0.62 |
| Modified Red Edge Simple Ratio Index | FC | -1e-4 | 2e-5 | 5e-10 | 0.61 |
| | | | | | |
| *Composition (Plant foliar traits):* | | | | | |
| Trait: Canopy nitrogen concentration | Intercept | -8e-3 | 1e-3 | 2e-7 | 0.45 |
| Trait: Carbon | FC | 3e-2 | 5e-3 | 5e-9 | 0.45 |
| Trait: Leaf Mass per Area (LMA) | Aspect | 3e-1 | 1e-1 | 6e-2 | 0.40 |
| | | | | | |
| *Function:* | | | | | |

| | | | | | |
|---|---|---|---|---|---|
| Evapotranspiration (nearest neighbour) | FC | -8e-3 | 1e-3 | 5e-16 | 0.55 |
| Evapotranspiration (statistical resampling) | FC | -3e-4 | 2e-3 | 8e-1 | 0.38 |

| Variable | **Ordered predictor terms (from most to least significant)** | | | | | |
|---|---|---|---|---|---|---|
| ***Structure:*** | | | | | | |
| **Canopy height** | Slope, FC | Intercept | Elevation | eCO$_2$ | Aspect | |
| **Biomass** | FC | Elevation | Slope | Intercept | eCO$_2$ | Aspect |
| ***Composition (Vegetation indices):*** | | | | | | |
| **Normalized Difference Vegetation Index (NDVI)** | eCO$_2$ | FC | Slope | Aspect | Elevation | Intercept |
| **Red Edge Normalized Difference VI** | eCO$_2$, FC | Aspect | Slope | Intercept | Elevation | |
| **Modified Red Edge Normalized Difference VI** | eCO$_2$ | FC, Elevation | Intercept | Aspect | Slope | |

| | | | | | |
|---|---|---|---|---|---|
| **Vogelmann Red Edge Index 1** | $eCO_2$, FC | Intercept | Aspect | Elevation | Slope |
| **Enhanced Vegetation Index** | $eCO_2$ | Intercept | Slope | Elevation, FC | Aspect |
| **Modified Red Edge Simple Ratio Index** | FC | $eCO_2$ | Intercept | Slope | Aspect | Elevation |

*Composition (Plant foliar traits):*

| | | | | | |
|---|---|---|---|---|---|
| **Trait: Canopy nitrogen concentration** | Intercept | $eCO_2$, Elevation | FC | Slope, Aspect | |
| **Trait: Carbon** | FC | Elevation | $eCO_2$ | Intercept | Slope | Aspect |
| **Trait: Leaf Mass per Area (LMA)** | Elevation | FC | Intercept | $eCO_2$, Slope | Aspect | |

*Function:*

| **Evapotranspiration (nearest neighbour)** | FC | eCO$_2$ | Elevation | Slope | Aspect | Intercept |
|---|---|---|---|---|---|---|
| **Evapotranspiration (statistical resampling)** | FC | Elevation | Slope | eCO$_2$ | Aspect | Intercept |

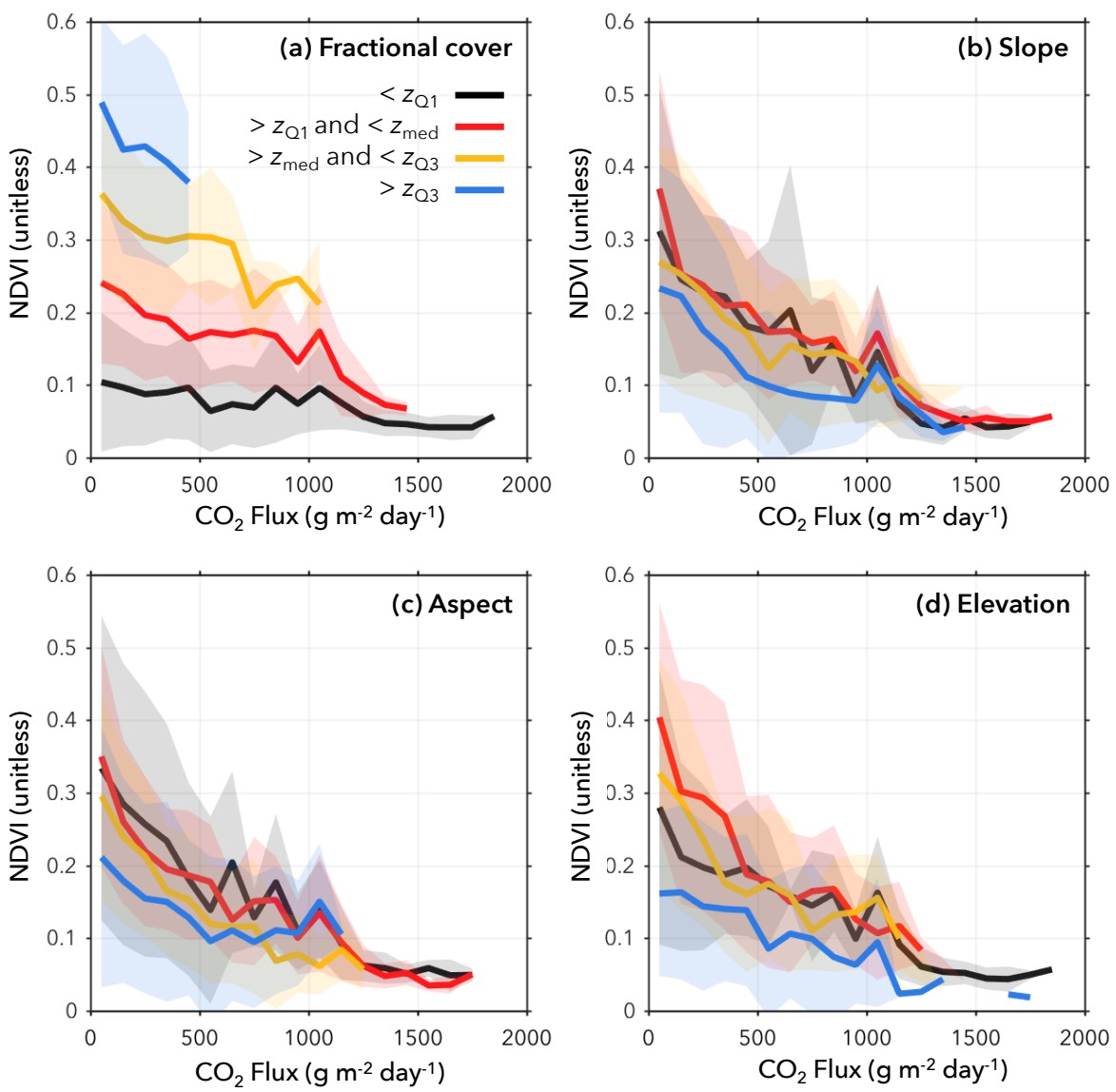

**Figure 3: Relationships between many ecological variables, including NDVI, and eCO₂ depend highly on confounding factors. The NDVI data is partitioned into quartiles and coloured such that, if z is the confounding variable (fractional cover, slope, aspect or elevation), then $z_{Q1}$ is the first quartile of the confounding data; $z_{med}$ is the median of the confounding data; and $z_{Q3}$ is the third quartile. Partitioning by fractional cover yields clear separations in the response variable (a) fractional cover, as expected, since rising eCO₂ will have a less measurable effect on sparse**

vegetation within the pixel. The impact of (b) slope, (c) elevation, and (d) aspect is less clear visually, but their contribution to the model is statistically significant.

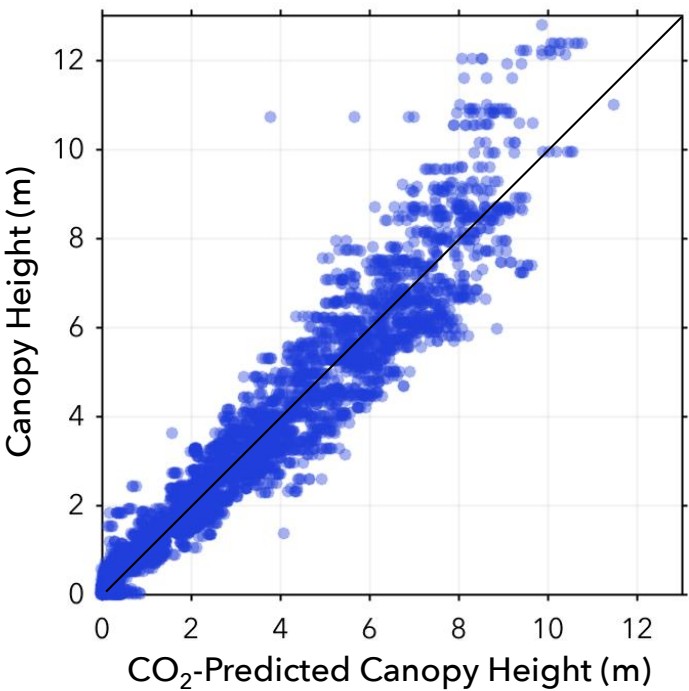

Figure 4: Canopy height is well modelled by the eCO$_2$ model, with an R$^2$=0.92, and the
5   1-1 line shown in black. However, the very tallest trees are not well captured.

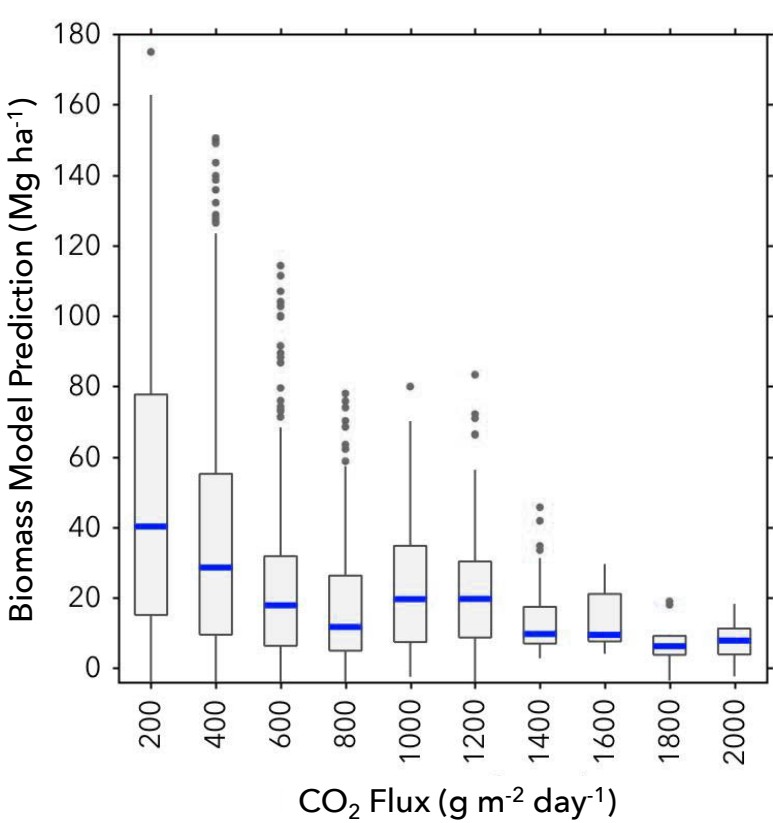

**Figure 5: The biomass model prediction is shown for increasing eCO₂. There is high variability at low eCO₂ values, but overall there is a small, but apparent, decrease in biomass with increasing eCO₂.**

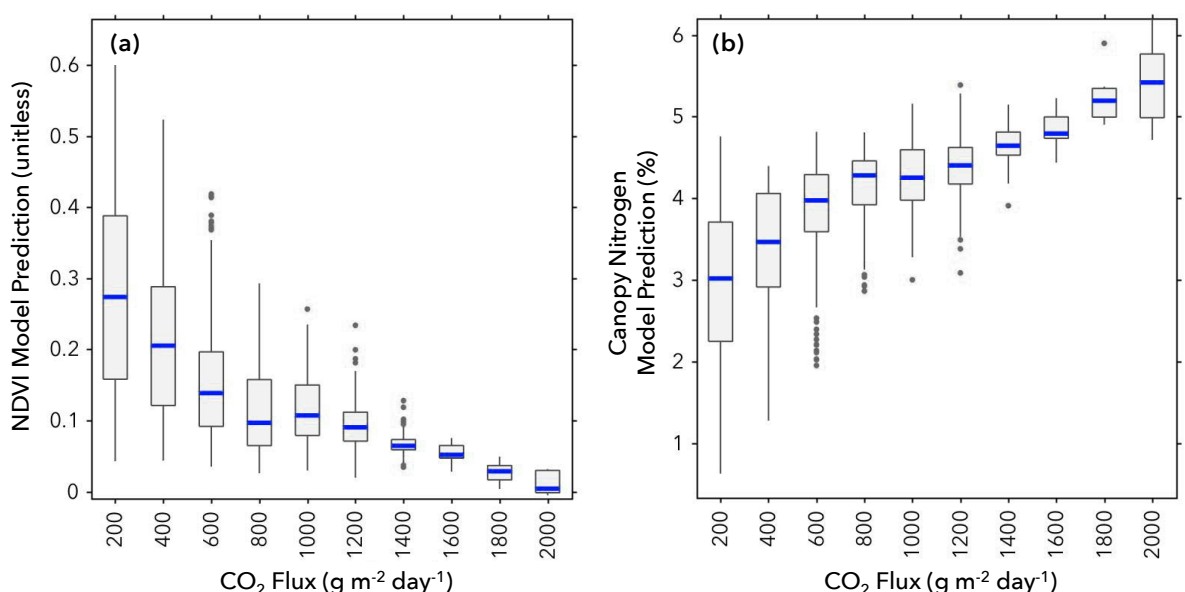

**Figure 6: (a) The modelled NDVI prediction is shown for predictor variable eCO₂. There is a decrease in NDVI for increasing eCO₂, despite larger variance at low eCO₂ values. (b) The modelled canopy nitrogen concentration trait prediction is shown for predictor variable eCO₂. There is a clear increase in canopy nitrogen concentration with increasing eCO₂.**

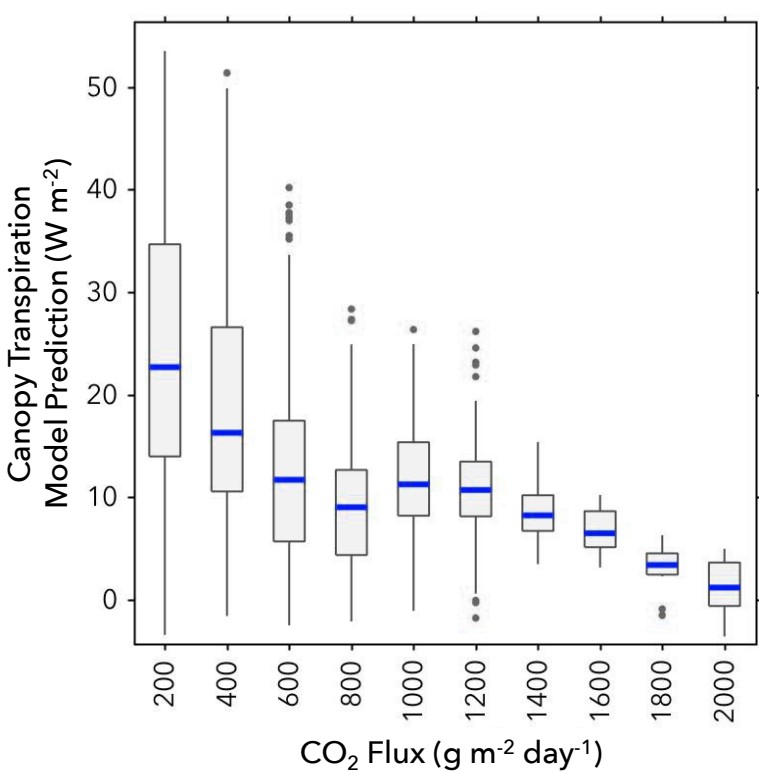

**Figure 7: The normalized canopy transpiration prediction is shown against predictor variable eCO₂, for training data with nearest neighbour resampling. There is a clear decrease in ET for increasing eCO₂, with larger variance at low eCO₂ values.**

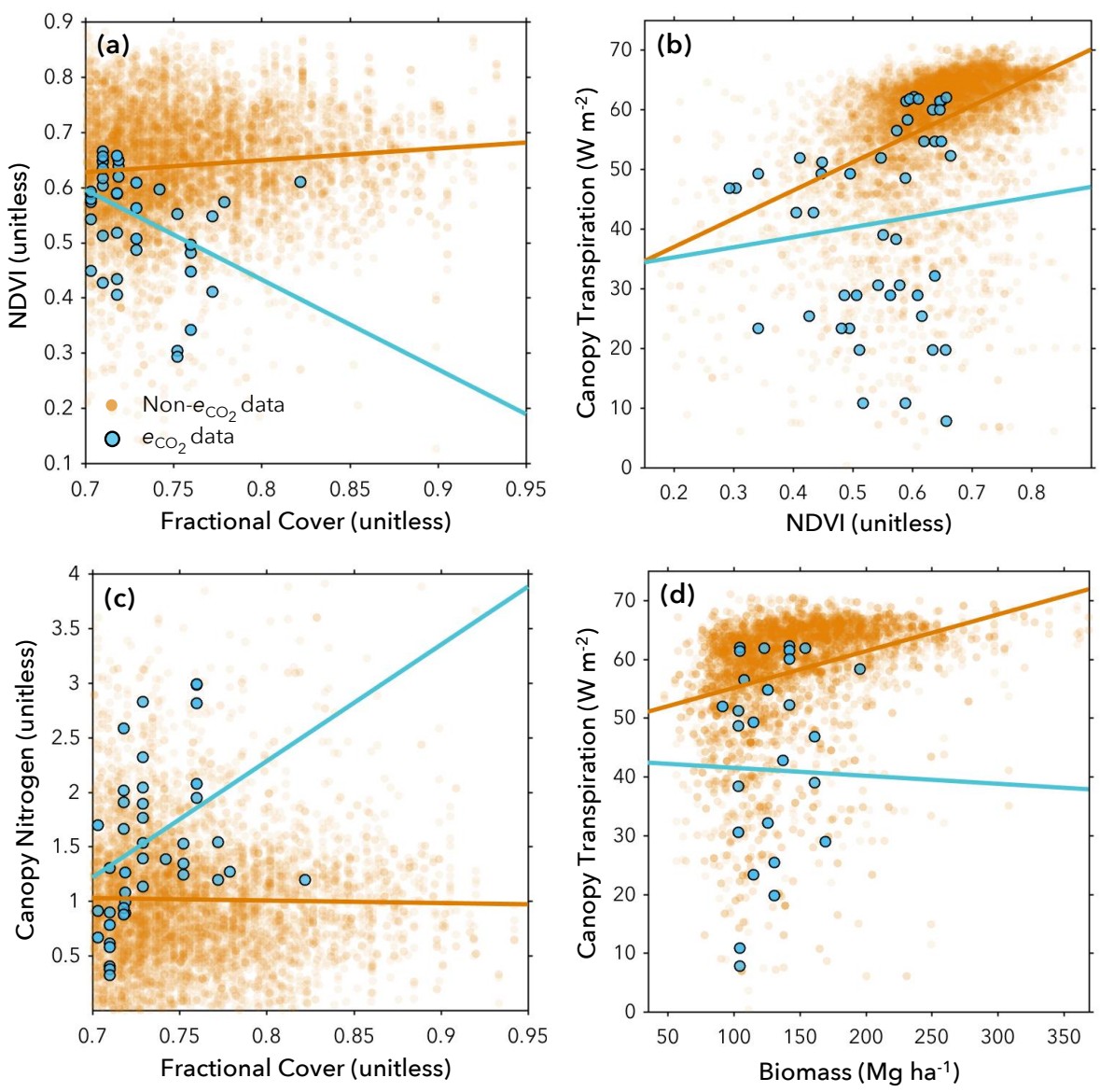

**Figure 8: Ecosystem dynamics inside (blue data points) and outside (orange data points) of the eCO₂ measurement boundaries contrast. (a) In the entire image, NDVI increases slightly with increasing fractional cover. In the small eCO₂ subset, NDVI appears to decrease with increasing fractional cover. (b) In the entire image, evapotranspiration increases with increasing NDVI, whereas the small eCO₂ subset seems to cover points with lower ET. (c) Across the entire image, the nitrogen trait remains constant with increasing fractional cover (thresholded at FC>0.7). In the small eCO₂ subset, the nitrogen trait appears to increase with increasing fractional cover. (d) In the entire scene, evapotranspiration increases with increasing biomass. In the small eCO₂ subset, the evapotranspiration seems to be lower, on average, for the same range of biomass values.**