# Peer review of "Ecosystem responses to elevated CO2 using airborne remote sensing at Mammoth Mountain, California"

_Biogeosciences, 2018_

## Short Comment (SC1) · 20 Apr 2018

Review: Ecosystem responses to elevated CO2 using airborne remote sensing at Mammoth Mountain, California

In this analysis Cawse-Nicholson use a volcanically active site where elevated CO2 fluxes have been monitored as a natural experiment to test vegetation response using remote sensing approaches. Given the contradictory results from previous studies at this site, it seems logical to revisit using new approaches. The rationale and methods for this study seemed logical and it provides a nice testing ground for testing a range of remote sensing techniques. I was quite surprised by the results showing the apparent suppression of growth (i.e. negative relationship between NDVI and soil CO2 flux), especially because this main conclusion was not clearly stated in the title or the abstract. It seems that the forests in this volcanic setting are responding adversely to something, but it is not clear why it would be elevated CO2 concentrations. I think that most folks reading the title, perhaps the abstract and looking at the figures will be a bit perplexed as I was. This is a really fascinating study system that is fairly complex in terms of terrain and gases emitted.

General Comments: The authors go to great lengths to control for distance from these hotspots of CO2 to derive a gradient over which to investigate vegetation responses, which is no easy task, especially using remotely derived metrics over complex terrain. In particular, I wonder how cold air drainage at night affects CO2 concentrations at these sights (Pypker et al. 2007). It is conceivable that much higher CO2 concentrations are found downslope than upslope or adjacent to these CO2 efflux hotspots (Fig. 2a). In fact, biomass hotspots appear to be adjacent or downslope from the CO2 hotspots (Fig. 2b); although it is difficult to discern without elevation contours.

Where on the A-Ci curve are we operating? The vegetation at these sites is responding to the partial pressure of CO2 in the atmosphere, among other gases at this site. Figure 1 suggests that the CO2 flux was maybe 2 orders of magnitude greater than typical estimates at non-volcanic sites (Jensen et al. 1996), but what is the partial pressure of CO2 in the atmosphere at these stites. I suspect that we are operating well above the asymptote on the A-Ci curve(Tissue, Griffin, and Ball 1999), such that we would see very little vegetation response to even large changes in the partial pressure of CO2.

What are the other gases are being emitted from this volcanic field? The negative relationship between CO2 soil flux and NDVI is perplexing and needs explaining. Are these particularly sulfur rich volcanic fields? Has anyone developed a 'rotten egg' remote sensing index? No but seriously, if there are significant sulfur emissions this could be leading to sulfuric acid deposition and cation loss from the soils, such that the negative response to soil fluxes could actually be the result of another gas that is
detrimental to plant growth other than CO2.

Specific Comments: The abstract is a bit vague reporting statistical relationships but not the apparent negative response to increased soil CO2 flux and without any response numbers (change in NDVI per change in Soil CO2 flux).

P2 L14 to 26 Perhaps the most fundamental flaw of FACE studies is very few have concomitant warming, which greatly limits our insight for the real world.

P3 What other gases are being emitted from these volcanic fields.

P3 L37 'can be applied'

P4 L20 is this g C or g CO2 per day...you might want to make this absolutely clear in the units

P4 L27 why were these data not just aggregated to a coarser resolution. Further smoothing of already smooth data may lead to loss of meaningful variance.

P5 L20 some discussion of cold air drainage important in this mountainous terrain (see Pypker below).

P7 L 12 as demonstrated by the authors- where?

P11 L 18 Why not use a random forest model to identify variables of greatest importance.

P12 L3 'well modeled' be more descriptive precisely or accurately?

Fig. 1 could benefit from a log y-scale or even better some estimate of pCO2

Fig. 3 the caption seems to be incomplete in describing all the panels.

References: Jensen, L. S., T. Mueller, K. R. Tate, D. J. Ross, J. Magid, and N. E. Nielsen. 1996. "Soil Surface CO2 Flux as an Index of Soil Respiration in Situ: A Comparison of Two Chamber Methods." Soil Biology & Biochemistry 28 (10): 1297–1306. Pypker, Thomas G., Michael H. Unsworth, Alan C. Mix, William Rugh, Troy Ocheltree,
Karrin Alstad, and Barbara J. Bond. 2007. "Using Nocturnal Cold Air Drainage Flow to Monitor Ecosystem Processes in Complex Terrain." Ecological Applications: A Publication of the Ecological Society of America 17 (3): 702–14. Tissue, David T., Kevin L. Griffin, and J. Timothy Ball. 1999. "Photosynthetic Adjustment in Field-Grown Ponderosa Pine Trees after Six Years of Exposure to Elevated CO2." Tree Physiology 19 (4-5). Oxford University Press: 221–28.

---

## Author Comment (AC1) · 26 Apr 2018

Review response to SC1

The authors thank Prof. Ballantyne for the positive review and useful feedback on this manuscript. This paper aimed to demonstrate the capability of both the natural elevated CO$_2$ experiment and the collection of airborne instruments to provide innovative ecology results.

We have responded to specific comments in red below:

Review: Ecosystem responses to elevated CO$_2$ using airborne remote sensing at

[Figure]

Mammoth Mountain, California

In this analysis Cawse-Nicholson use a volcanically active site where elevated $CO_2$ fluxes have been monitored as a natural experiment to test vegetation response using remote sensing approaches. Given the contradictory results from previous studies at this site, it seems logical to revisit using new approaches. The rationale and methods for this study seemed logical and it provides a nice testing ground for testing a range of remote sensing techniques. I was quite surprised by the results showing the apparent suppression of growth (i.e. negative relationship between NDVI and soil $CO_2$ flux), especially because this main conclusion was not clearly stated in the title or the abstract. It seems that the forests in this volcanic setting are responding adversely to something, but it is not clear why it would be elevated $CO_2$ concentrations. I think that most folks reading the title, perhaps the abstract and looking at the figures will be a bit perplexed as I was. This is a really fascinating study system that is fairly complex in terms of terrain and gases emitted.

We thank you for noting the innovativeness on using volcanically-derived elevated $CO_2$ as a means to assess long term ecosystem responses through remote sensing approaches. Some of the results were indeed unexpected — but, this is exactly why such a study is needed. It may be possible that the NDVI decrease is due to a progressive nutrient limitation, as has been suggested throughout the literature, but has never been tested empirically. However, much more in depth investigation is required to determine the underlying mechanisms explaining the results. As such, we frame this paper as more suggestive than conclusive, ideally leading to further work on this topic.

General Comments: The authors go to great lengths to control for distance from these hotspots of $CO_2$ to derive a gradient over which to investigate vegetation responses, which is no easy task, especially using remotely derived metrics over complex terrain. In particular, I wonder how cold air drainage at night affects $CO_2$ concentrations at these sights (Pypker et al. 2007). It is conceivable that much higher $CO_2$ concentrations are found downslope than upslope or adjacent to these $CO_2$ efflux hotspots

[Figure]

(Fig. 2a). In fact, biomass hotspots appear to be adjacent or downslope from the $CO_2$ hotspots (Fig. 2b); although it is difficult to discern without elevation contours.

We agree that a more thorough assessment of $CO_2$ flow through the landscape is needed. It is remarkable that we were able to detect clear signals from soil fluxes alone; we expect that the results would be improved with above- and within-canopy $CO_2$ measurements, and better tracking over time. Given the available measurements from USGS, the best we could do was shift the ground $CO_2$ dataset in all cardinal directions, to see if this resulted in an improved fit. The best model fit was found at the original ground $CO_2$ location. We will include elevation contours in the revised manuscript.

Where on the A-Ci curve are we operating? The vegetation at these sites is responding to the partial pressure of $CO_2$ in the atmosphere, among other gases at this site. Figure 1 suggests that the $CO_2$ flux was maybe 2 orders of magnitude greater than typical estimates at non-volcanic sites (Jensen et al. 1996), but what is the partial pressure of $CO_2$ in the atmosphere at these sites. I suspect that we are operating well above the asymptote on the A-Ci curve(Tissue, Griffin, and Ball 1999), such that we would see very little vegetation response to even large changes in the partial pressure of $CO_2$.

The partial pressures at Mammoth are about 60% of sea level. The fact that we see systematic ecosystem effects suggests that elevation is not on the flat part of the A-Ci curve. In other words, even if elevation were to reduce the $CO_2$ effect, we still are seeing strong $CO_2$ effects regardless, highlighting just how important and strong of a response we are able to detect. We will add this discussion to the revised manuscript.

What are the other gases are being emitted from this volcanic field? The negative relationship between $CO_2$ soil flux and NDVI is perplexing and needs explaining. Are these particularly sulfur rich volcanic fields? Has anyone developed a 'rotten egg' remote sensing index? No but seriously, if there are significant sulfur emissions this could be leading to sulfuric acid deposition and cation loss from the soils, such that

the negative response to soil fluxes could actually be the result of another gas that is detrimental to plant growth other than $CO_2$.

There is no significant $H_2S$ nor any $SO_2$ present at soil levels at this site; see, for example, data in Sorey et al 1998, Werner et al, 2014, and a number of papers on volcanic degassing at Mammoth Mountain by our USGS co-author Lewicki (2006, 2007, 2008, 2012, 2014). Furthermore, the direct areas of $CO_2$ emissions, which impacts the local soil conditions, do not contain any vegetation (kill-zones) and were removed when we used a threshold for fractional cover. We will add and clarify this detail in the revised manuscript.

Specific Comments: The abstract is a bit vague reporting statistical relationships but not the apparent negative response to increased soil $CO_2$ flux and without any response numbers (change in NDVI per change in Soil $CO_2$ flux).

We will add more statistics to the revised manuscript.

P2 L14 to 26 Perhaps the most fundamental flaw of FACE studies is very few have concomitant warming, which greatly limits ourinsight for the real world.

The FACE studies have been invaluable to our understanding of the $CO_2$ effect, which contributes to among the largest uncertainties in projections of Earth's climate. While it is true that they primarily assess $CO_2$, we argue that the actual biggest limitation of FACE is the short durations —there has been no way to assess long-term changes in ecosystems. This is where the long term emissions of volcanic $CO_2$ can play a game changing role in how to assess the long term $CO_2$ effect on ecosystems.

P3 What other gases are being emitted from these volcanic fields.

As discussed above, $CO_2$ dominates by up to 99% of gas volume.

P3 L37 'can be applied'

This will be corrected.

P4 L20 is this g C or g $CO_2$ per day...you might want to make this absolutely clear in the units

These are $g.m^{-2}.d^{-1}$ of $CO_2$, and will be clarified.

P4 L27 why were these data not just aggregated to a coarser resolution. Further smoothing of already smooth data may lead to loss of meaningful variance.

The original raw field $CO_2$ flux measurement data are no longer available. We worked from the 1m data that were provided to us by the USGS, which are aggregates of data collected by several different surveys in the 2011–2012 time frame, with the Horseshoe Lake area visited multiple times to characterize any very subtle temporal variation (Fig. 1 in Werner et al, 2014).

P5 L20 some discussion of cold air drainage important in this mountainous terrain (see Pypker below).

Thank you, we will include this discussion and reference.

P7 L 12 as demonstrated by the authors- where?

As demonstrated by Ma et al (2018). This will be made clearer in the next version.

P11 L 18 Why not use a random forest model to identify variables of greatest importance.

We considered random forest models and obtained similar result. We presented the results of the linear regression since the model itself is more easily interpretable by the reader.

P12 L3 'well modeled' be more descriptive precisely or accurately?

Canopy height and biomass were accurately modelled with high $R^2$. Will edit in the revision.

Fig. 1 could benefit from a log y-scale or even better some estimate of $pCO_2$

This will be modified in the revised manuscript.

Fig. 3 the caption seems to be incomplete in describing all the panels.

This will be modified in the revised manuscript.

References:

Jensen, L. S., T. Mueller, K. R. Tate, D. J. Ross, J. Magid, and N. E. Nielsen. 1996. "Soil Surface CO2 Flux as an Index of Soil Respiration in Situ: A Comparison of Two Chamber Methods." Soil Biology Biochemistry 28 (10): 1297–1306.

Pypker, Thomas G., Michael H. Unsworth, Alan C. Mix, William Rugh, Troy Ocheltree, Karrin Alstad, and Barbara J. Bond. 2007. "Using Nocturnal Cold Air Drainage Flow to Monitor Ecosystem Processes in Complex Terrain." Ecological Applications: A Publication of the Ecological Society of America 17 (3): 702–14.

Tissue, David T., Kevin L. Griffin, and J. Timothy Ball. 1999. "Photosynthetic Adjustment in Field-Grown Pon- derosa Pine Trees after Six Years of Exposure to Elevated CO2." Tree Physiology 19 (4-5). Oxford University Press: 221–28.

References (by authors):

Ma, P., Kang, E. L., Braverman, A., and Nguyen, H, 2018. Spatial statistical downscaling for constructing high resolution nature runs in global observing system simulation experiments. In review.

Sorey, M.L., Evans, W.C., Kennedy, B.M., Farrar, C.D., Hainsworth, L.J. and Hausback, B., 1998. Carbon dioxide and helium emissions from a reservoir of magmatic gas beneath Mammoth Mountain, California. Journal of Geophysical Research: Solid Earth, 103(B7), pp.15303-15323.

Werner, C., Bergfeld, D., Farrar, C.D., Doukas, M.P., Kelly, P.J. and Kern, C., 2014. Decadal-scale variability of diffuse CO2 emissions and seismicity revealed from long-term monitoring (1995–2013) at Mammoth Mountain, California, USA. Journal of Vol-

canology and Geothermal Research, 289, pp.51-63.

Lewicki, J.L., Hilley, G.E., Tosha, T., Aoyagi, R., Yamamoto, K. and Benson, S.M., 2006. Dynamic coupling of volcanic CO2 flow and wind at the HorseshoeLake tree kill, Mammoth Mountain, CA. Geophysical Research Letters, 34(LBNL–62375).

Lewicki, J.L., Hilley, G.E., Tosha, T., Aoyagi, R., Yamamoto, K. and Benson, S.M., 2007. Dynamic coupling of volcanic CO2 flow and wind at the Horseshoe Lake tree kill, Mammoth Mountain, California. Geophysical Research Letters, 34(3), L03401. DOI: 10.1029/2006GL028848

Lewicki, J.L., Fischer, M.L. and Hilley, G.E., 2008. Six-week time series of eddy covariance CO2 flux at Mammoth Mountain, California: performance evaluation and role of meteorological forcing. Journal of Volcanology and Geothermal Research, 171(3-4), pp.178-190.

Lewicki, J.L., Hilley, G.E., Dobeck, L. and Marino, B.D., 2012. Eddy covariance imaging of diffuse volcanic CO2 emissions at Mammoth Mountain, CA, USA. Bulletin of volcanology, 74(1), pp.135-141.

Lewicki, J.L. and Hilley, G.E., 2014. Multi-scale observations of the variability of magmatic CO2 emissions, Mammoth Mountain, CA, USA. Journal of Volcanology and Geothermal Research, 284, pp.1-15.

---

## Referee Comment (RC1) · Anonymous Referee #1 · 2 Jun 2018

Overview and significance

In this analysis Cawse-Nicholson et al. describe ecological attributes measured through several remote sensing platforms in relation to ground-measured and modeled elevated CO2 originating from volcanic degassing. The primary objective and novelty of this study is to estimate the impact of elevated CO2 on plant growth and whole ecosystems by utilization naturally occurring gradients of elevated CO2 from volcanic degassing. Previous experiments and studies in estimating the impact of elevated CO2 on plants and ecosystems approach scaling limitations; whether through limited species diversity, space or time of exposure to elevated CO2, and/or cost of artificially

elevating CO2. Therefore conclusions of experimental CO2 enhancements are limited to relatively few species and over short periods of time without leveraging natural gradients of elevated CO2. Methodologies to use natural CO2 gradients in determining plant and ecosystem responses to elevated CO2 described herein, in conjunction with elevated CO2 experiments, will fill important gaps in understand how individual plants to whole ecosystems will respond to continually increasing levels of CO2. The hope for the methodology described herein is for it to be applied where gradients of CO2 exists in order to understand the impact of elevated CO2 across multiple biomes.

General comments:

The authors outline their objectives as

1. Evaluate the viability of using a passively degassing volcano system to study the properties of ecosystesm; 2. assess the detectability of ecological responses to elevated soil CO2 emissions via airborne data alone; 3. Present key lessons enabling future studies to extend our framework to other biomes.

Objective 1 is approached using soil CO2 flux measurements at a spatial resolution of 1 meter. This was made possible through the records of soil CO2 flux measurements at Mammoth Mountain. The authors acknowledge that measurements from soil CO2 fluxes will be much different and more stable than atmospheric fluxes of CO2 (page 5 line 10 and page 15 line 35). This approach makes estimating actual atmospheric CO2 measurements intractable under known methodologies but is strong enough to infer that atmospheric CO2 was greater than background where soil CO2 flux was greater.

Mammoth Mountain included a tree-kill zone for which the authors selected the trees around this zone. The presence of a tree-kill zone naturally leads to hypotheses that elevated CO2 will have a negative effects on vegetation at some point up the CO2 gradient. Previous studies pointing this out are cited in the manuscript and detected by NDVI (Rouse et al. 2010 and Cholathat et al. 2011) and through tree ring analysis (Biondi and Fessenden 1999). The authors addition to these cited analyses to include vegetation indices from AVIRIS and biomass measurements derived from Lidar as proposed in Objective 2. Soil CO2 flux was shown to be a significant predictor for these indices and remotely sensed attributes. While the vegetation indices are all slightly different they are largely related to one another vs. the other measurements of biomass, plant foliar traits, and canopy evapotranspiration. Some explanation as to why looking at several different vegetation indices and comparing each individually to enhanced CO2 may be beneficial for understanding how plant physiology is impacted and what methodologies may be selected in investigating other biomes (Objective 3).

The hypothesis and observations that elevated CO2 has negative effects on vegetation is contrary to many greenhouse and FACE experiments of artificially enhancing CO2, but is likely related to the intensity of elevated CO2 at the volcanic site. The authors also speculate that elevated soil CO2 may lead to oxygen deprivation of roots and soil acidification (page 15 line 34 and cited in Farrar et al., 1995; Qi et al., 1994; McGee and Gerlach, 1998). This has major confounding effects on being able to use volcanic degassing to detect the impact of elevated atmospheric CO2 on photosynthesis and carbon sequestration if suitable soil chemistry for plant growth becomes a limiting factor. Rouse et al. (2010) did observe that in multispectral analysis of vegetation revealed that plant vigor degraded under high CO2 but slightly increased under low CO2. Along the same lines that Cawse-Nicholson et al. have speculated, slight increase in plant vigor may exist in zones where soil O2 is still above a certain threshold and/or soils are adequately buffered. I suggest that in order for the methodology put forth by Cawse-Nicholson et al. to effectively capture the impact of elevated atmospheric CO2 on ecosystem traits that measurements be made of soil O2, soil pH, and atmospheric CO2 be made in future studies. As is, the study of Cawse-Nicholson et al. provides a valuable step forward in being able to scale-up the impact of elevated CO2 on plants to whole ecosystems and across differing biomes.

Specific comments: - Table 2. As the primary subject of this paper is elevated CO2 a

complete ranking of the explanatory variables against CO2 would be informative even for dependent variables in which eCO2 was not the most influential variable.

Technical corrections: Page 11 line 15 slope and aspect seem mixed up as slopes of 350 are not feasible.

---

## Referee Comment (RC2) · Anonymous Referee #2 · 18 Jun 2018

This is an interesting study using a purported natural CO2 enhancement gradient to understand ecosystem scale responses to elevated CO2. The authors use a linear regression model to control for a couple of covariates to discern the effect of eCO2 on structure and process.

Overall, the empirical model results in confusing results, which the authors try to explain by referring to similar studies in other naturally enhanced CO2 systems. I find the discussion quite speculative and have two concerns on the study and the usefulness of volcanic-CO2 seepage as an experimental setting.

1) The authors argue that the Mammoth Mt region is very well studied and that variability in CO2 over time and space is minimal, and that the ecosystems in the area are in some equilibrium with the seepage. But even ignoring variability before measurements began, the Figure 1 shows very high variability since measurements first started. I don't think we can say with any confidence what the CO2 exposure has been over time and space, and whether the current study reflects the equilibrium conditions to eCO2. 2) The authors focus only on eCO2 as a driver of variability in structure and processes. Soil conditions (physical and chemical) are overlooked and it is quite possible that some sort of chemical toxicity is interacting with plant growth and causing the unusual 'eCO2 responses' that the team finds.

Minor comments: - Define MASTER and ASO when first used - Effect of canopy height model (selecting tallest pixel in each 1 m2 grid cell) will likely bias the biomass estimate to outliers, why not use percentiles, i.e. 90th, to avoid this artefact? - Please discuss a bit more the sample size used to develop the plant traits models with AVIRIS.

---

## Author Comment (AC2) · 27 Jul 2018

Overview and significance In this analysis Cawse-Nicholson et al. describe ecological attributes measured through several remote sensing platforms in relation to ground-measured and modeled elevated CO2 originating from volcanic degassing. The primary objective and novelty of this study is to estimate the impact of elevated CO2 on plant growth and whole ecosystems by utilization naturally occurring gradients of elevated CO2 from volcanic degassing. Previous experiments and studies in estimating the impact of elevated CO2 on plants and ecosystems approach scaling limitations; whether through limited species diversity, space or time of exposure to elevated CO2, and/or cost of artificially elevating CO2. Therefore conclusions of experimental CO2 enhancements are limited to relatively few species and over short periods of time without leveraging natural gradients of elevated CO2. Methodologies to use natural CO2 gradients in determining plant and ecosystem responses to elevated CO2 described herein, in conjunction with elevated CO2 experiments, will fill important gaps in understand how individual plants to whole ecosystems will respond to continually increasing levels of CO2. The hope for the methodology described herein is for it to be applied where gradients of CO2 exists in order to understand the impact of elevated CO2 across multiple biomes.

We thank the reviewer for noting the novelty of our study in overcoming scaling limitations of previous studies, and the important gap that we aim to fill in understanding how plants and ecosystems will respond to continually rising CO2.

General comments: The authors outline their objectives as 1. Evaluate the viability of using a passively degassing volcano system to study the properties of ecosystems; 2. assess the detectability of ecological responses to elevated soil CO2 emissions via airborne data alone; 3. Present key lessons enabling future studies to extend our framework to other biomes.

Objective 1 is approached using soil CO2 flux measurements at a spatial resolution of 1 meter. This was made possible through the records of soil CO2 flux measurements at Mammoth Mountain. The authors acknowledge that measurements from soil CO2 fluxes will be much different and more stable than atmospheric fluxes of CO2 (page 5 line 10 and page 15 line 35). This approach makes estimating actual atmospheric CO2 measurements intractable under known methodologies but is strong enough to infer that atmospheric CO2 was greater than background where soil CO2 flux was greater.

Mammoth Mountain included a tree-kill zone for which the authors selected the trees around this zone. The presence of a tree-kill zone naturally leads to hypotheses that elevated CO2 will have a negative effects on vegetation at some point up the CO2 gradient. Previous studies pointing this out are cited in the manuscript and detected by NDVI (Rouse et al. 2010 and Cholathat et al. 2011) and through tree ring anal- and biomass measurements derived from Lidar as proposed in Objective 2. Soil CO2 flux was shown to be a significant predictor for these indices and remotely sensed attributes. While the vegetation indices are all slightly different they are largely related to one another vs. the other measurements of biomass, plant foliar traits, and canopy evapotranspiration. Some explanation as to why looking at several different vegetation indices and comparing each individually to enhanced CO2 may be beneficial for understanding how plant physiology is impacted and what methodologies may be selected in investigating other biomes (Objective 3).

While all vegetation indices are indeed related, they differ enough to be considered independent variables. E.g. some account for soil moisture, others weigh plant greenness more heavily. This was an exploratory effort in investigating the effects of CO2 on any measure of plant function, composition, and structure, and so we attempted to cover all avenues of investigation. A note to this effect will be included in the next revision of the manuscript.

We note for clarification that the "kill-zone" is the exact location of volcanic gas seeps along fractures, where CO2 is predominantly emitted from the soil—a property of the soil being altered by the emission; but, we focus on the "fertilization zone", which is away from those emission points, with unaffected soils, where tree canopies are exposed to the CO2, which has diffused in the atmosphere away from the emission points.

The hypothesis and observations that elevated CO2 has negative effects on vegetation is contrary to many greenhouse and FACE experiments of artificially enhancing CO2, but is likely related to the intensity of elevated CO2 at the volcanic site. The authors

also speculate that elevated soil CO2 may lead to oxygen deprivation of roots and soil acidification (page 15 line 34 and cited in Farrar et al., 1995; Qi et al., 1994; McGee and Gerlach, 1998). This has major confounding effects on being able to use volcanic degassing to detect the impact of elevated atmospheric CO2 on photosynthesis and carbon sequestration if suitable soil chemistry for plant growth becomes a limiting factor. Rouse et al. (2010) did observe that in multispectral analysis of vegetation revealed that plant vigor degraded under high CO2 but slightly increased under low CO2. Along the same lines that Cawse-Nicholson et al. have speculated, slight increase in plant vigor may exist in zones where soil O2 is still above a certain threshold and/or soils are adequately buffered. I suggest that in order for the methodology put forth by Cawse-Nicholson et al. to effectively capture the impact of elevated atmospheric CO2 on ecosystem traits that measurements be made of soil O2, soil pH, and atmospheric CO2 be made in future studies. As is, the study of Cawse-Nicholson et al. provides a valuable step forward in being able to scale-up the impact of elevated CO2 on plants to whole ecosystems and across differing biomes.

We thank the reviewer for complementing our study as a valuable step forward, as well as the suggestion for measurements in future studies. As one of our objectives was to provide guidance for future studies, these suggestions fit well with our objectives.

As in our previous response above, we will clarify that any vegetation impacts are due not to soil changes from direct CO2 emissions, as we excluded the emission zones from our study. We will also clarify that the effects should not necessarily be given a subjective description of 'negative'; rather, it is important to note that the CO2 fertilization effect is unlikely to continue indefinitely, particularly at the same rates that FACE studies have shown only in the short-term. All other experiments have been unable to show long-term effects. Our study suggests that over the scale of decades, some of these hypothesized greening or biomass increases may not be sustainable. Other results, such as an increase in canopy nitrogen with increasing CO2, do seem to remain consistent with our study, however.

Specific comments: - Table 2. As the primary subject of this paper is elevated CO2, a complete ranking of the explanatory variables against CO2 would be informative even for dependent variables in which eCO2 was not the most influential variable.

This is a good suggestion, and the complete ranking will be included.

Technical corrections: Page 11 line 15 slope and aspect seem mixed up as slopes of 350 are not feasible.

Thank you. This has been corrected.

---

## Author Comment (AC3) · 28 Jul 2018

This is an interesting study using a purported natural CO2 enhancement gradient to understand ecosystem scale responses to elevated CO2. The authors use a linear regression model to control for a couple of covariates to discern the effect of eCO2 on structure and process.

We thank the reviewer for noting the interest of our study using a well-documented natural CO2 enhancement to understand ecosystem scale responses to elevated CO2.

Overall, the empirical model results in confusing results, which the authors try to explain

**Printer-friendly version**

by referring to similar studies in other naturally enhanced CO2 systems.

Some of the results of this natural long-term exposure experiment may seem confusing at first because they contradict shorter-term experiments. But, this is exactly why we did the study—if we knew what the results were going to be, there would be no reason for the study. Moreover, the results highlight numerous points made throughout the literature with respect to the FACE experiments—their short-term nature has been unable to uncover long-term results, which is exactly the unique strength and a primary purpose of our study. We edited the manuscript to make these points more clear (and to make understanding the results less confusing).

I find the discussion quite speculative and have two concerns on the study and the usefulness of volcanic-CO2 seepage as an experimental setting.

We agree that the Discussion is structured more as a Discussion, less as Results. We tried to make clear that this study was exploratory, rather than definitive, and that this study was meant to identify both potential signals as well as design elements for further study.

1) The authors argue that the Mammoth Mt region is very well studied and that variability in CO2 over time and space is minimal, and that the ecosystems in the area are in some equilibrium with the seepage. But even ignoring variability before measurements began, the Figure 1 shows very high variability since measurements first started. I don't think we can say with any confidence what the CO2 exposure has been over time and space, and whether the current study reflects the equilibrium conditions to eCO2.

It is a fundamental principle of volcanology that all active volcanoes emit CO2 continuously during their entire life cycle. The CO2 emissions at Mammoth Mountain have been well known since at least 1989, and their variability well documented by repeated CO2 efflux mapping between at least 1995 (ongoing), by the USGS (Werner et al. 2014). Multiple sites have since at least 1995 been continuously monitored for CO2 by the USGS (McGee Gerlach, 1998), showing highly invariant continuous excess CO2 BGD
emissions at these sites (e.g., Rogie et al 2001; Werner et al., 2014). The CO2 seeps at the site have been known for even longer (Varekamp Buseck 1984, and geothermal assessment reports from the 1970s at least). All these studies show that the active Mammoth Mountain volcanic system has experienced a replenishment of the magmatic CO2 source in the deep subsurface in about 1989, possibly already an earlier one in 1978, though no systematic CO2 measurements were conducted in that earlier time period (Hill, 1996). Werner et al 2014) show remarkable spatial consistency for 9 years of systematic measurements at the CO2 gas seeps on Mammoth Mountain.

2) The authors focus only on eCO2 as a driver of variability in structure and processes. Soil conditions (physical and chemical) are overlooked and it is quite possible that some sort of chemical toxicity is interacting with plant growth and causing the unusual 'eCO2 responses' that the team finds.

The reviewer is correct in that soil chemistry is altered at the points of CO2 emission (e.g., McGee Gerlach, 1998). However, we excluded those areas, instead focusing on the fertilization zone, which is away from those emission points, with unaffected soils, where tree canopies are exposed to the CO2, which has diffused in the atmosphere away from the emission points.

Minor comments: - Define MASTER and ASO when first used - Effect of canopy height model (selecting tallest pixel in each 1 m2 grid cell) will likely bias the biomass estimate to outliers, why not use percentiles, i.e. 90th, to avoid this artefact? - Please discuss a bit more the sample size used to develop the plant traits models with AVIRIS.

We will define MASTER and ASO at first use. Outliers have already been removed as part of the preprocessing of the biomass estimate. We will clarify this point. We will include information on the foliar trait model development in the next version of the manuscript.

Hill, D.P., 1996. Earthquakes and carbon dioxide beneath Mammoth Mountain, California. Seismological Research Letters, 67(1), pp.8-15.

BGD
McGee, K.A. and Gerlach, T.M., 1998. Annual cycle of magmatic CO2 in a tree-kill soil at Mammoth Mountain, California: Implications for soil acidification. Geology, 26(5), pp.463-466.

Rogie, J.D., Kerrick, D.M., Sorey, M.L., Chiodini, G. and Galloway, D.L., 2001. Dynamics of carbon dioxide emission at Mammoth Mountain, California. Earth and Planetary Science Letters, 188(3-4), pp.535-541.

Sorey, M.L., Farrar, C.D., Evans, W.C., Hill, D.P., Bailey, R.A., Hendley, J.W. and Stauffer, P.H., 1996. Invisible CO2 gas killing trees at Mammoth Mountain, California. US Geological Survey Fact Sheet, pp.172-96. URL: https://pubs.usgs.gov/fs/fs172-96/

J.C. Varekamp, P.R. Buseck (1984): Changing mercury anomalies in Long Valley, California: Indication for magma movement or seismic activity. Geology ; 12 (5): 283–286

Werner, C., Bergfeld, D., Farrar, C.D., Doukas, M.P., Kelly, P.J. and Kern, C., 2014. Decadal-scale variability of diffuse CO2 emissions and seismicity revealed from long-term monitoring (1995–2013) at Mammoth Mountain, California, USA. Journal of Volcanology and Geothermal Research, 289, pp.51-63.

BGD

---

## Author Response (AR2)

**Response to Associate Editor**

**Dear Dr. Schöngart,**

5 We thank you and the reviewers for your responses and valuable suggestions. We have made small modifications to the manuscript throughout, in order to clarify issues, and further discussions. In particular, we have included more technical details on the model output, and have included discussions on other factors that may affect the results we have found (particularly the decrease in NDVI).

**10**

With increasing CO2, we found a decrease in ET and an increase in canopy nitrogen, both consistent with theory, suggesting more water and nutrient use efficient canopies. However, we also observed a decrease in NDVI with increasing CO2, which may be consistent with increased efficiency of fewer leaves.

**15**

Thank you for your continued consideration.

**Best,**

Kerry Cawse-Nicholson

20

I have read the three reviews as well as your responses and how you will address those in the revised version of bg-2018-73. All reviewers acknowledge the importance of this study and its novelty filling an important gap in understanding how plants and ecosystems will

25 respond to continually rising CO2 coming up with different results as experiments, such as FACE, limited in space and time. I will ask you to address the valuable comments made by the reviewers and decided to accept bg-2018-73 for publication after a minor revision.

Dr. A. P. Ballantyne notes the importance of this study in testing different remote-sensing

- 30 techniques to test vegetation responses to elevated CO2 concentrations at the long-term. He focuses in his review much on the rather unexpected negative relationship between NDVI and soil CO2 flux which is an important observation. Possible factors, e.g., nutrient limitation, extreme climate events, oxygen deprivation of roots, soil acidification and plant vigor (see also comments of both anonymous referees) should be discussed indicating the
- 35 need for further studies in this complex system.

We thank the reviewers for raising all of these potential factors, and we have included a paragraph summarising these at the end of the Discussions section, as well as a more detailed analysis of several factors that have been repeatedly raised (such as soil acidification) discussed earlier in the section. Of particular interest is the nutrient limitation

5 – it is a plausible scenario that NDVI decreases because the leaves are more nutritionally efficient; thus there are fewer leaves needed. This is discussed in more detail in the manuscript.

The author's should also address the influence of elevation reducing the CO2 effect.

- 10 We have discussed both partial pressures and cold air drainage. The partial pressures at Mammoth are about 60% of sea level. The fact that we see systematic ecosystem effects suggests that elevation is not on the flat part of the A-Ci curve. In other words, even if elevation were to reduce the CO2 effect, we still are seeing strong CO2 effects regardless, highlighting just how important and strong of a response we are able to detect. We will add
- 15 this discussion to the revised manuscript.

Please also indicate some additional statistics and the order of magnitude of change in NDVI per change in Soil CO2 flux in the abstract.

Additional statistics have been provided throughout the manuscript, and we have included 20 the change in NDVI per change in soil CO2 flux in the abstract. I have used the 200 – 800

g/m2/day range in the abstract, since population sizes decrease beyond that level.

I encourage the authors also to indicate the limitations of the FACE experiments around the world in contrast to the long-term emissions of volcanic CO2 emissions especially focusing

- 25 on the warming trend. The FACE studies have been invaluable to our understanding of the CO2 effect, which contributes to among the largest uncertainties in projections of Earth's climate. While it is true that they primarily assess CO2, we argue that the actual biggest limitation of FACE is the short durations—there has been no way to assess long-term changes in ecosystems.
- 30 This is where the long term emissions of volcanic CO2 can play a game changing role in how to assess the long term CO2 effect on ecosystems.

Even if 99% of the emitted gas volume is CO2, the author should mention what other gases are emitted with the respective numbers. Even if CO2 is a trace gas in the atmosphere it has

35 a tremendous impact on the Earth system!

There is no significant H2S nor any SO2 present at soil levels at this site; see, for example, data in Sorey et al 1998, Werner et al, 2014, and a number of papers on volcanic degassing at Mammoth Mountain by our USGS co-author Lewicki (2006, 2007, 2008, 2012, 2014). The other contributing gases by volume are N2 and O2.

5

Please do also consider the minor concerns indicated my Dr. Ballantyne. We have clarified many points throughout the manuscript in response to Dr. Ballantyne's suggestions.

- 10 Anonymous Referee #1 notes the novelty of this study filling an important gap in understanding how plants and ecosystems will respond to continually rising CO2 by overcoming limitations of previous studies in spatial and temporal scales. As he indicated, the authors should indicate some explanation to why looking at several different vegetation indices and comparing each individually to enhanced CO2 may be beneficial for
- 15 understanding how plant physiology and growth is impacted and what methodologies may be selected in investigating other ecosystems. While all vegetation indices are indeed related, they differ enough to be considered independent variables. E.g. some account for soil moisture, others weight plant greenness more heavily. This was an exploratory effort in investigating the effects of CO2 on any
- 20 measure of plant function, composition, and structure, and so we attempted to cover all avenues of investigation. This discussion has been included in the newest revision of the manuscript.

As Dr. Ballantyne, anonymous reviewer #1 also focuses a lot on possible negative impacts (oxygen deprivation of roots, soil acidification, plant vigor) on elevated CO2 (contrasting results obtained by the FACE experiment) indicating the need for further studies especially on soil pH and O2 and citing important literature which should be considered by the authors.

We note for clarification that the "kill-zone" is the exact location where  $CO_2$  is emitted from

- 30 the soil—a property of the soil being altered by the emission; but, we focus on the "fertilization zone", which is away from those emission points, with unaffected soils, where tree canopies are exposed to the CO2, which has diffused in the atmosphere away from the emission points. We have also included a discussion on other negative impacts, as above.
- 35 The strength of the paper is the long-term monitoring vs. short-term observations as in FACE, and this should be evidenced in the discussion (as pointed out by the other reviews).

**A more detailed discussion of the long-term monitoring aspect has been included.**

Please consider also the minor concerns and technical corrections indicated by reviewer #1. We have addressed the minor concerns raised by this reviewer.

5

Anonymous referee #2 acknowledges the long-term character of this monitoring, but also criticizes the empirical model used for the analysis of many covariates (which are closely related to each other as pointed out by referee #1) to discern the effects of eCO2 on structure and dynamic.

10

This reviewer has two major concerns. The first focuses on the variability of eCO2 over time and the authors give an interesting response indicating the history of measurements of CO2 emissions at Mammouth Mountain and I would find in interesting to include this in the Material and Method section. The second one was addresses soil conditions as important

- 15 factor on structure and dynamics of vegetation which was already indicated by the other two reviewers and should be sufficiently addressed in the revised version of bg-2018-73. We have included the review response in the revised manuscript, and have addressed soil conditions as indicated previously.
- 20 Please also consider the minor concerns indicated by referee #2. We have addressed the minor concerns raised by this reviewer.

I am looking forward to the revised version of bg-2018-73. Thank you for choosing Biogeosciences to publish this fascinating study.

25 Thank you!

**Anonymous Referee #1**

Received and published: 2 June 2018

Overview and significance

In this analysis Cawse-Nicholson et al. describe ecological attributes measured through 5 several remote sensing platforms in relation to ground-measured and modeled elevated CO2 originating from volcanic degassing. The primary objective and novelty of this study is to estimate the impact of elevated CO2 on plant growth and whole ecosystems by utilization naturally occurring gradients of elevated CO2 from volcanic degassing. Previous experiments and studies in estimating the impact of elevated CO2 on plants and ecosystems approach

- 10 scaling limitations; whether through limited species diversity, space or time of exposure to elevated CO2, and/or cost of artificially elevating CO2. Therefore conclusions of experimental CO2 enhancements are limited to relatively few species and over short periods of time without leveraging natural gradients of elevated CO2. Methodologies to use natural CO2 gradients in determining plant and ecosystem responses to elevated CO2 described herein, in
- 15 conjunction with elevated CO2 experiments, will fill important gaps in understand how individual plants to whole ecosystems will respond to continually increasing levels of CO2. The hope for the methodology described herein is for it to be applied where gradients of CO2 exists in order to understand the impact of elevated CO2 across multiple biomes.

We thank the reviewer for noting the novelty of our study in overcoming scaling limitations of previous studies, and the important gap that we aim to fill in understanding how plants and ecosystems will respond to continually rising CO2.

General comments:

The authors outline their objectives as

Evaluate the viability of using a passively degassing volcano system to study the properties
 of ecosystems; 2. assess the detectability of ecological responses to elevated soil CO2 emissions via airborne data alone; 3. Present key lessons enabling future studies to extend our framework to other biomes.

Objective 1 is approached using soil CO2 flux measurements at a spatial resolution of 1 meter. This was made possible through the records of soil CO2 flux measurements at Mammoth Mountain. The authors acknowledge that measurements from soil CO2 fluxes will be much different and more stable than atmospheric fluxes of CO2 (page 5 line 10 and page 15 line

5 35). This approach makes estimating actual atmospheric CO2 measurements intractable under known methodologies but is strong enough to infer that atmospheric CO2 was greater than background where soil CO2 flux was greater.

Mammoth Mountain included a tree-kill zone for which the authors selected the trees around this zone. The presence of a tree-kill zone naturally leads to hypotheses that elevated CO2

- 10 will have a negative effects on vegetation at some point up the CO2 gradient. Previous studies pointing this out are cited in the manuscript and detected by NDVI (Rouse et al. 2010 and Cholathat et al. 2011) and through tree ring anal- and biomass measurements derived from Lidar as proposed in Objective 2. Soil CO2 flux was shown to be a significant predictor for these indices and remotely sensed attributes. While the vegetation indices are all slightly
- 15 different they are largely related to one another vs. the other measurements of biomass, plant foliar traits, and canopy evapotranspiration. Some explanation as to why looking at several different vegetation indices and comparing each individually to enhanced CO2 may be beneficial for understanding how plant physiology is impacted and what methodologies may be selected in investigating other biomes (Objective 3).
- 20 While all vegetation indices are indeed related, they differ enough to be considered independent variables. E.g. some account for soil moisture, others weight plant greenness more heavily. This was an exploratory effort in investigating the effects of CO2 on any measure of plant function, composition, and structure, and so we attempted to cover all avenues of investigation. A note to this effect will be included in the next revision of the manuscript.
- 25 We note for clarification that the "kill-zone" is the exact location where CO2 is emitted from the soil—a property of the soil being altered by the emission; but, we focus on the "fertilization zone", which is away from those emission points, with unaffected soils, where tree canopies are exposed to the CO2, which has diffused in the atmosphere away from the emission points.

The hypothesis and observations that elevated CO2 has negative effects on vegetation is 30 contrary to many greenhouse and FACE experiments of artificially enhancing CO2, but is likely related to the intensity of elevated CO2 at the volcanic site. The authors also speculate that elevated soil CO2 may lead to oxygen deprivation of roots and soil acidification (page 15 line 34 and cited in Farrar et al., 1995; Qi et al., 1994; McGee and Gerlach, 1998). This has major confounding effects on being able to use volcanic degassing to detect the impact of elevated atmospheric CO2 on photosynthesis and carbon sequestration if suitable soil chemistry for plant growth becomes a limiting factor. Rouse et al. (2010) did observe that in multispectral analysis of vegetation revealed that plant vigor degraded under high CO2 but slightly

- 5 increased under low CO2. Along the same lines that Cawse-Nicholson et al. have speculated, slight increase in plant vigor may exist in zones where soil O2 is still above a certain threshold and/or soils are adequately buffered. I suggest that in order for the methodology put forth by Cawse-Nicholson et al. to effectively capture the impact of elevated atmospheric CO2 on ecosystem traits that measurements be made of soil O2, soil pH, and atmospheric CO2 be
- 10 made in future studies. As is, the study of Cawse-Nicholson et al. provides a valuable step forward in being able to scale-up the impact of elevated CO2 on plants to whole ecosystems and across differing biomes.

We thank the reviewer for complimenting our study as a valuable step forward, as well as the suggestion for measurements in future studies. As one of our objectives was to provide guidance for future studies, these suggestions fit well with our objectives.

As in our previous response above, we will clarify that any vegetation impacts are due not to soil changes from direct CO2 emissions, as we excluded the emission zones from our study. We will also clarify that the effects should not necessarily be given a subjective description of 'negative'; rather, it is important to note that the CO2 fertilization effect is unlikely to continue indefinitely, particularly at the same rates that FACE studies have shown only in the short-term. All other experiments have been unable to show long-term effects. Our study suggests that over the scale of decades, some of these hypothesized greening or biomass increases may not be sustainable. Other results, such as an increase in canopy nitrogen with increasing 25 CO2, do seem to remain consistent with our study, however.

Specific comments: - Table 2. As the primary subject of this paper is elevated  $CO_2$ , a complete ranking of the explanatory variables against CO2 would be informative even for dependent variables in which eCO2 was not the most influential variable.

This is a good suggestion, and the complete ranking will be included.

Technical corrections: Page 11 line 15 slope and aspect seem mixed up as slopes of 350 are not feasible.

Thank you. This has been corrected.

**Review response to SC1**

5

The authors thank Prof. Ballantyne for the positive review and useful feedback on this manuscript. This paper aimed to demonstrate the capability of both the natural elevated CO2 experiment and the collection of airborne instruments to provide innovative ecology results.

We have responded to specific comments in red below:

Review: Ecosystem responses to elevated CO2 using airborne remote sensing at Mammoth 10 Mountain, California

In this analysis Cawse-Nicholson use a volcanically active site where elevated CO2 fluxes have been monitored as a natural experiment to test vegetation response using remote sensing approaches. Given the contradictory results from previous studies at this site, it seems logical

- 15 to revisit using new approaches. The rationale and methods for this study seemed logical and it provides a nice testing ground for testing a range of remote sensing techniques. I was quite surprised by the results showing the apparent suppression of growth (i.e. negative relationship between NDVI and soil CO2 flux), especially because this main conclusion was not clearly stated in the title or the abstract. It seems that the forests in this volcanic setting are
- 20 responding adversely to something, but it is not clear why it would be elevated CO2 concentrations. I think that most folks reading the title, perhaps the abstract and looking at the figures will be a bit perplexed as I was. This is a really fascinating study system that is fairly complex in terms of terrain and gases emitted.

We thank you for noting the innovativeness on using volcanically-derived elevated CO2 as a

- 25 means to assess long term ecosystem responses through remote sensing approaches. Some of the results were indeed unexpected—but, this is exactly why such a study is needed. It may be possible that the NDVI decrease is due to a progressive nutrient limitation, as has been suggested throughout the literature, but has never been tested empirically. However, much more in depth investigation is required to determine the underlying mechanisms explaining
- 30 the results. As such, we frame this paper as more suggestive than conclusive, ideally leading to further work on this topic.

General Comments: The authors go to great lengths to control for distance from these hotspots of CO2 to derive a gradient over which to investigate vegetation responses, which is

35 no easy task, especially using remotely derived metrics over complex terrain. In particular, I wonder how cold air drainage at night affects CO2 concentrations at these sights (Pypker et al. 2007). It is conceivable that much higher CO2 concentrations are found downslope than upslope or adjacent to these CO2 efflux hotspots (Fig. 2a). In fact, biomass hotspots appear to

be adjacent or downslope from the CO2 hotspots (Fig. 2b); although it is difficult to discern without elevation contours.

We agree that a more thorough assessment of  $CO_2$  flow through the landscape is needed. It is remarkable that we were able to detect clear signals from soil fluxes alone; we expect that the

5 results would be improved with above- and within-canopy CO2 measurements, and better tracking over time. Given the available measurements from USGS, the best we could do was shift the ground CO2 dataset in all cardinal directions, to see if this resulted in an improved fit. The best model fit was found at the original ground CO2 location. We will include elevation contours in the revised manuscript.

10

Where on the A-Ci curve are we operating? The vegetation at these sites is responding to the partial pressure of CO2 in the atmosphere, among other gases at this site. Figure 1 suggests that the CO2 flux was maybe 2 orders of magnitude greater than typical estimates at non-volcanic sites (Jensen et al. 1996), but what is the partial pressure of CO2 in the atmosphere

15 at these sites. I suspect that we are operating well above the asymptote on the A-Ci curve(Tissue, Griffin, and Ball 1999), such that we would see very little vegetation response to even large changes in the partial pressure of CO2. The partial pressures at Mammoth are about 60% of sea level. The fact that we see systematic

ecosystem effects suggests that elevation is not on the flat part of the A-Ci curve. In other

20 words, even if elevation were to reduce the CO2 effect, we still are seeing strong CO2 effects regardless, highlighting just how important and strong of a response we are able to detect. We will add this discussion to the revised manuscript.

What are the other gases are being emitted from this volcanic field? The negative relationship

- 25 between CO2 soil flux and NDVI is perplexing and needs explaining. Are these particularly sulfur rich volcanic fields? Has anyone developed a 'rotten egg' remote sensing index? No but seriously, if there are significant sulfur emissions this could be leading to sulfuric acid deposition and cation loss from the soils, such that the negative response to soil fluxes could actually be the result of another gas that is detrimental to plant growth other than CO2.
- 30 There is no significant H2S nor any SO2 present at soil levels at this site; see, for example, data in Sorey et al 1998, Werner et al, 2014, and a number of papers on volcanic degassing at Mammoth Mountain by our USGS co-author Lewicki (2006, 2007, 2008, 2012, 2014). Furthermore, we excluded the direct areas of CO2 emissions, which impacts the local soil conditions, not representative of the larger ecosystem. We will add and clarify this detail in 25 the revised measurement.
- 35 the revised manuscript.

Specific Comments: The abstract is a bit vague reporting statistical relationships but not the apparent negative response to increased soil CO2 flux and without any response numbers (change in NDVI per change in Soil CO2 flux).

We will add more statistics to the revised manuscript.

5

P2 L14 to 26 Perhaps the most fundamental flaw of FACE studies is very few have concomitant warming, which greatly limits ourinsight for the real world. The FACE studies have been invaluable to our understanding of the CO2 effect, which contributes to among the largest uncertainties in projections of Earth's climate. While it is

10 true that they primarily assess  $CO_2$ , we argue that the actual biggest limitation of FACE is the short durations—there has been no way to assess long-term changes in ecosystems. This is where the long term emissions of volcanic  $CO_2$  can play a game changing role in how to assess the long term  $CO_2$  effect on ecosystems.

P3 What other gases are being emitted from these volcanic fields.

15 As discussed above, CO2 dominates by up to 99% of gas volume.P3 L37 'can be applied' This will be corrected.

P4 L20 is this g C or g CO2 per day...you might want to make this absolutely clear in the units

- These are g.m-2.d-1 of CO2, and will be clarified.
   P4 L27 why were these data not just aggregated to a coarser resolution. Further smoothing of already smooth data may lead to loss of meaningful variance.
   The original raw field CO2 flux measurement data were not available anymore. We worked from the 1m data that were provided to us by the USGS, which are aggregates of data
- 25 collected by several different surveys in the 2011-2012 time frame, with the Horseshoe Lake area visited multiple times to characterize any very subtle temporal variation (Fig. 1 in Werner et al, 2014).

P5 L20 some discussion of cold air drainage important in this mountainous terrain (see Pypker below).

- 30 Thank you, we will include this discussion and reference.
  P7 L 12 as demonstrated by the authors- where?
  As demonstrated by Ma et al (2018). This will be made clearer in the next version.
  P11 L 18 Why not use a random forest model to identify variables of greatest importance.
  We considered random forest models and obtained similar result. We presented the results of
  the linear regression since the model itself is more assily interpretable by the reader.
- the linear regression since the model itself is more easily interpretable by the reader.
   P12 L3 'well modeled' be more descriptive precisely or accurately?
   Canopy height and biomass were accurately modelled with high R2. Will edit in the revision.
   Fig. 1 could benefit from a log y-scale or even better some estimate of pCO2

This will be modified in the revised manuscript. Fig. 3 the caption seems to be incomplete in describing all the panels. This will be modified in the revised manuscript.

- 5 References: Jensen, L. S., T. Mueller, K. R. Tate, D. J. Ross, J. Magid, and N. E. Nielsen. 1996. "Soil Surface CO2 Flux as an Index of Soil Respiration in Situ: A Com- parison of Two Chamber Methods." Soil Biology & Biochemistry 28 (10): 1297–1306. Pypker, Thomas G., Michael H. Unsworth, Alan C. Mix, William Rugh, Troy Ocheltree, Karrin Alstad, and Barbara J. Bond. 2007. "Using Nocturnal Cold Air Drainage Flow to Monitor Ecosystem
- 10 Processes in Complex Terrain." Ecological Applications: A Pub- lication of the Ecological Society of America 17 (3): 702–14. Tissue, David T., Kevin L. Griffin, and J. Timothy Ball. 1999. "Photosynthetic Adjustment in Field-Grown Ponderosa Pine Trees after Six Years of Exposure to Elevated CO2." Tree Physiology 19 (4-5). Oxford University Press: 221–28.

**15 **References (by authors):**

Ma, P., Kang, E. L., Braverman, A., and Nguyen, H, 2018. Spatial statistical downscaling for constructing high resolution nature runs in global observing system simulation experiments. In review.

Sorey, M.L., Evans, W.C., Kennedy, B.M., Farrar, C.D., Hainsworth, L.J. and Hausback, B.,

20 1998. Carbon dioxide and helium emissions from a reservoir of magmatic gas beneath Mammoth Mountain, California. Journal of Geophysical Research: Solid Earth, 103(B7), pp.15303-15323.

Werner, C., Bergfeld, D., Farrar, C.D., Doukas, M.P., Kelly, P.J. and Kern, C., 2014. Decadal-scale variability of diffuse CO2 emissions and seismicity revealed from long-

- 25 term monitoring (1995–2013) at Mammoth Mountain, California, USA. Journal of Volcanology and Geothermal Research, 289, pp.51-63.
  - Lewicki, J.L., Hilley, G.E., Tosha, T., Aoyagi, R., Yamamoto, K. and Benson, S.M., 2006. Dynamic coupling of volcanic CO2 flow and wind at the HorseshoeLake tree kill, Mammoth Mountain, CA. Geophysical Research Letters, 34(LBNL--62375).
- 30 Lewicki, J.L., Hilley, G.E., Tosha, T., Aoyagi, R., Yamamoto, K. and Benson, S.M., 2007. Dynamic coupling of volcanic CO2 flow and wind at the Horseshoe Lake tree kill, Mammoth Mountain, California. Geophysical Research Letters, 34(3), L03401. DOI: 10.1029/2006GL028848

Lewicki, J.L., Fischer, M.L. and Hilley, G.E., 2008. Six-week time series of eddy covariance

35 CO2 flux at Mammoth Mountain, California: performance evaluation and role of meteorological forcing. Journal of Volcanology and Geothermal Research, 171(3-4), pp.178-190.

- Lewicki, J.L., Hilley, G.E., Dobeck, L. and Marino, B.D., 2012. Eddy covariance imaging of diffuse volcanic CO2 emissions at Mammoth Mountain, CA, USA. Bulletin of volcanology, 74(1), pp.135-141.
- Lewicki, J.L. and Hilley, G.E., 2014. Multi-scale observations of the variability of magmatic 5 CO2 emissions, Mammoth Mountain, CA, USA. Journal of Volcanology and
  - Geothermal Research, 284, pp.1-15.

**Anonymous Referee #3**

This is an interesting study using a purported natural CO2 enhancement gradient to understand ecosystem scale responses to elevated CO2. The authors use a linear regression model to control for a couple of covariates to discern the effect of eCO2 on structure and 5 process.

We thank the reviewer for noting the interest of our study using a well-documented natural  $CO_2$  enhancement to understand ecosystem scale responses to elevated  $CO_2$ .

Overall, the empirical model results in confusing results, which the authors try to explain by referring to similar studies in other naturally enhanced CO2 systems.

- 10 Some of the results may seem confusing because they go against shorter-term experiments. But, this is exactly why we did the study—if we knew what the results were going to be, there would be no reason for the study. Moreover, the results highlight numerous points made throughout the literature with respect to the FACE experiments—their short-term nature has been unable to uncover long-term results, which is exactly a primary purpose of our study.
- 15 We edited the manuscript to make these points more clear (and to make understanding the results less confusing).

I find the discussion quite speculative and have two concerns on the study and the usefulness of volcanic-CO2 seepage as an experimental setting.

We agree that the Discussion is structured more as a Discussion, less as Results. We tried to 20 make clear that this study was exploratory, rather than definitive, and that this study was meant to identify both potential signals as well as design elements for further study.

1) The authors argue that the Mammoth Mt region is very well studied and that variability in CO2 over time and space is minimal, and that the ecosystems in the area are in some equilibrium with the seepage. But even ignoring variability before measurements began, the

25 Figure 1 shows very high variability since measurements first started. I don't think we can say with any confidence what the CO2 exposure has been over time and space, and whether the current study reflects the equilibrium conditions to eCO2.

It is a fundamental principal of volcanology that all active volcanoes emit  $CO_2$  continuously during their entire life cycle. The  $CO_2$  emissions at Mammoth Mountain have been well

known since at least 1989, and their variability well documented by repeated CO2 efflux mapping between at least 1995 (ongoing), by the USGS (Werner et al. 2014). The CO2 seeps at the site have been known for even longer (Varekamp & Buseck 1984, and geothermal assessment reports from the 1970s at least). All these studies show that the active Mammoth

5 Mountain volcanic system has experienced a replenishment of the magmatic CO2 source in the deep subsurface in about 1989, possibly already an earlier one in 1978, though no systematic CO2 measurements were conducted in that earlier time period (Hill, 1996). Werner et al show remarkable spatial consistency for 9 years of systematic measurements at the CO2 gas seeps on Mammoth Mountain.

10

2) The authors focus only on eCO2 as a driver of variability in structure and processes. Soil conditions (physical and chemical) are overlooked and it is quite possible that some sort of chemical toxicity is interacting with plant growth and causing the unusual 'eCO2 responses' that the team finds.

- 15 The reviewer is correct in that soil chemistry is altered at the points of CO2 emission. However, we excluded those areas, instead focusing on the fertilization zone, which is away from those emission points, with unaffected soils, where tree canopies are exposed to the CO2, which has diffused in the atmosphere away from the emission points.
- 20 Minor comments: Define MASTER and ASO when first used Effect of canopy height model (selecting tallest pixel in each 1 m2 grid cell) will likely bias the biomass estimate to outliers, why not use percentiles, i.e. 90th, to avoid this artefact? Please discuss a bit more the sample size used to develop the plant traits models with AVIRIS.

We will define MASTER and ASO at first use. Outliers have already been removed as part of the preprocessing of the biomass estimate. We will clarify this point.

We will include information on the foliar trait model development in the next version of the manuscript.

Hill, D.P., 1996. Earthquakes and carbon dioxide beneath Mammoth Mountain, California. Seismological Research Letters, 67(1), pp.8-15.

McGee, K.A. and Gerlach, T.M., 1998. Annual cycle of magmatic CO2 in a tree-kill soil at Mammoth Mountain, California: Implications for soil acidification. Geology, 26(5), pp.463-466.

Sorey, M.L., Farrar, C.D., Evans, W.C., Hill, D.P., Bailey, R.A., Hendley, J.W. and Stauffer, P.H.,
1996. Invisible CO2 gas killing trees at Mammoth Mountain, California. US Geological Survey Fact Sheet, pp.172-96. URL: https://pubs.usgs.gov/fs/fs172-96/

J.C. Varekamp, P.R. Buseck (1984): Changing mercury anomalies in Long Valley, California: Indication for magma movement or seismic activity. *Geology* ; 12 (5): 283–286

Werner, C., Bergfeld, D., Farrar, C.D., Doukas, M.P., Kelly, P.J. and Kern, C., 2014. Decadal scale variability of diffuse CO2 emissions and seismicity revealed from long-term monitoring (1995–2013) at Mammoth Mountain, California, USA. Journal of Volcanology and Geothermal Research, 289, pp.51-63.

**Ecosystem responses to elevated CO2 using airborne remote sensing at Mammoth Mountain, California**

K-Kerry Cawse-Nicholson1, J-Joshua B. Fisher1, C-Caroline A. Famiglietti1, A-Amy Braverman1, F-Florian M. Schwandner1,2, J-Jennifer L. Lewicki3, P-Philip A. Townsend4,

5 D.David S. Schimel1, R.Ryan Pavlick1, K.Kathryn J. Bormann1, A.Antonio Ferraz1, E.Emily L. Kang5, P.Pulong Ma5, R.Robert R. Bogue1, T.Thomas Youmans1, D.David C. Pieri1

1Jet Propulsion Laboratory, California Institute of Technology, Pasadena, CA, USA 2Joint Institute for Regional Earth System Science & and Engineering, University of California Los Angeles, Los Angeles, CA, USA

3United States Geological Survey, Menlo Park, CA, USA
 4University of Wisconsin-Madison, Madison, WI, USA
 5University of Cincinnati, Cincinnati, OH, USA

Correspondence to: Kerry Cawse-Nicholson, kcawseni@jpl.nasa.gov © 2018. All rights reserved

- 15 **Abstract.** We present an exploratory study examining the use of airborne remote sensing observations to detect ecological responses to elevated CO2 emissions from active volcanic systems. To evaluate these ecosystem responses, existing spectroscopic, thermal, and lidar data acquired over forest ecosystems on Mammoth Mountain volcano, California, were exploited, along with *in situ* measurements of persistent volcanic soil CO2 fluxes. The
- 20 elevated CO2 response was used to statistically model ecosystem structure, composition and function, evaluated via data products including biomass, plant foliar traits and vegetation indices, and evapotranspiration (ET). Using regression ensemble models, we found that soil CO2 flux was a significant predictor for ecological variables, including canopy greenness (Normalized Vegetation Difference Index, -(NDVI7), canopy nitrogen, ET, and biomass. With
- 25 increasing CO2, we found a decrease in ET and an increase in canopy nitrogen, both consistent with theory, suggesting more water and nutrient use efficient canopies. However, we also observed a decrease in NDVI with increasing CO2 (a mean NDVI of 0.27 at 200 g m2 day1 CO2 reduced to a mean NDVI of 0.10 at 800 g m2 day1 CO2). This is inconsistent with theory; though consistent with increased efficiency of fewer leaves<del>nonetheless, with</del>
- 30 more efficient leaves, it may be that the trees needed fewer leaves. We found no changea decrease in aboveground biomass with increasing CO2, also inconsistent with theory; but, we did findalso found a decrease in biomass variance, pointing to a long-term homogenization of structure with elevated CO2, saw decreasing NDVI, increasing ET, increasing canopy nitrogen, and a decrease in biomass variance, pointing to trees with fewer, but more efficient
- 35 leaves. Additionally, the relationships between ecological variables changed with increasingly elevated (volcanically influenced) over non-volcanic "background" soil CO2

**Formatted: Subscript**

| Formatted: Superscrip | ot       |
|-----------------------|----------|
| Formatted: Superscrip | ot       |
| Formatted: Subscript  |          |
| Formatted: Superscrip | ot       |
| Formatted: Superscrip | ot       |
| Formatted: Subscript  |          |
| Formatted: Font color | : Text 1 |
| Formatted: Font color | : Text 1 |

fluxes, suggesting a shift in coupling/decoupling among ecosystem structure, composition, and function synergies. For example, ET and biomass were significantly correlated for areas without elevated  $CO_2$  flux, but decoupled with elevated  $CO_2$  flux. This study demonstrates that a) volcanic systems show great potential as a means to study the properties of ecosystems

5 and their responses to elevated CO2 emissions and b) these ecosystem responses are measureablemeasurable using a suite of airborne remotely sensed data.

**1** Introduction**

Terrestrial ecosystems have consistently taken up carbon over the past century, in excess or balancing losses due to deforestation and land use change, and this sink has grown with time (Le Quéré et al., 2016; Schimel et al., 2015). Much debate, however, has centred on the

- 5 drivers of this uptake. Suggested mechanisms include nitrogen deposition (Peterson & and Melillo, 1985), land use (Schimel, 1995), and the direct effects of carbon dioxide on plant growth (Norby et al., 2016). The last, which proposes that increased atmospheric CO2 yields increased photosynthetic rates, is both the most probable and the most controversial. Although a multitude of experiments have shown positive photosynthetic responses to
- 10 increased CO2 consistent with the observed growth in the terrestrial sink (Drake et al., 1997), many ecologists have argued that plant growth in intact ecosystems is limited by water, light or nutrients, rather than CO2 (Körner, 2006; McGuire et al., 1995).

The Free-Air Carbon Enrichment (FACE) experiments, introduced in the 1990s, allow for

- 15 CO2 fertilization of intact ecosystems by creating controlled fumigation conditions without the use of a growth chamber (Lewin et al., 1994). The FACE studies have been invaluable to our understanding of the CO2 effect, which contributes to among the largest uncertainties in projections of Earth's climate. These studies have shown some consistent responses indicative of enhanced growth (Norby et al., 2016), as well as other physiological,
- 20 morphological and ecosystem consequences, but also suffer from several structural deficiencieslimitations. Perhaps most notably, only short-term study periods are feasible; the longest-running experiment spanned only a decade, while atmospheric CO2 has been steadily rising for more than an order of magnitude longer than that duration. FACE can thus elucidate physiological responses to elevated CO2, but cannot unshroudand it is difficult to
- 25 measure slower processes like plant acclimation, shifts in species dominance induced by CO2, or other long-term mechanisms mediated by changes to soil organic matter and nutrients. Additionally, because FACE experiments are vastly expensive to construct and operate, they tend to be small in scale, limited in replicability, and homogeneous in species, soils and landscapes.
- 30

The FACE studies have been invaluable to our understanding of the CO2-effect, which contributes to among the largest uncertainties in projections of Earth's climate. While it is true that they primarily assess CO2, we argue that the actual biggest limitation of FACE is the short durations — there has been no way to assess long term changes in ecosystems. This is

35 where the long-term localized emissions of volcanic CO2 can play a game changing role in how to assess the long-term CO2 effect on ecosystems.

Field Code Changed Field Code Changed Field Code Changed

Field Code Changed

**Field Code Changed**

**Field Code Changed**

**Field Code Changed**

[revised manuscript text omitted]

Mammoth Mmountain is an upper montane forest ecosystem, characterised by abundant *Pinus contorta* (lodgepole pine7), and also by mature stands of *Abies magnifica* (red fir7).
30 *Pinus jeffreyi* (Jeffrey pine7), *Pinus albicaulis* (whitebark pine7), and *Juniperus occidentalis* (western juniper) (Potter, 1998). The elevation of our study areas ranged from 2700 to 2950 m. Tree-kill soils are immature High Sierra soils formed from granite, pumice, rhyolite, and obsidian parent materials (McGee & and Gerlach, 1998).

**2.1.1 Ground measurements**

We investigated soil CO2 fluxes within five actively degassing areas on Mammoth Mountain documented by Werner et al. (2014) in 2011 and 2012, which represents a period of relatively high emissions. (up to 2000 g m-2 day-1 of CO2). As described by Werner et al. (2014), fluxes

- 5 were measured along fixed grid points using the accumulation chamber method (Rahn et al., 1996). In situ measurements were obtained using a West Systems® (Florence, Italy) portable fluxmeter equipped with a LI-COR820 infrared gas analyzeranalyser. Based on statistical analysis, Werner et al. (2014) found soil CO2 fluxes measured within areas of volcanic CO2 emissions to be significantly elevated over background areas that were dominated by soil
- 10 respiration of CO2. Maps of soil CO2 flux were simulated from in-situ measurements at 1 m resolution using a sequential Gaussian simulation algorithm by these authors and we resampled their data to the Airborne Visible/Infrared Imaging Spectrometer (AVIRIS) resolution (13 m) using nearest neighbourneighbour resampling. Conventionally, studies of diffuse soil degassing of CO2 on volcanoes have emphasized understanding of the modes,
- 15 locations, geometries, and changes in volcanic flank degassing for purposes of volcanologicalyulcanological research, hazard assessment, and monitoring. In many cases, volcanologists have focussed on areas associated with sufficient emissions of heat and CO2 that vegetation has been killed off. In this study however, we focussed on vegetated areas where somewhat more mildly enhanced levels of volcanic CO2 emissions into the forest
- 20 ecosystems might be beneficial for plant growth, rather than adverse. As such, we investigated zones and gradients *around* tree-kill areas, excluding areas that were barren or contained dead trees by filtering by fractional vegetation cover, where appropriate. The tree-kill areas have local soil conditions that are not representative of the larger ecosystem. In addition, because tree-kill areas on Mammoth Mountain are largely associated with "cold"
- CO2 emissions, we completely avoided confounding influences of hydrothermal heat or acidic vapour emission on ecosystem response. Indeed, tThere is no significant H2S nor any SO2 present at soil levels at this site, and CO2 makes up ~99% of the gas by volume; see, for example, data in (Sorey et al 1998, Werner et al, 2014), and a number of papers on volcanic degassing at Mammoth Mountain (Lewicki 2006, 2007, 2008, 2012, 2014). The remaining
   1% is made up of N2 and O2.

The use of a high-spatial-resolution time-averaged (to limit the influence of varying meteorological conditions) map of canopy-level atmospheric CO2 concentration would be most applicable to assess ecosystem response to elevated atmospheric CO2 concentrations.

35 However, such maps are unavailable. We therefore took advantage of the extensive record of soil CO2 fluxes available for Mammoth Mountain. Although the effects of elevated CO2 in the soil may be difficult to de-convolve from elevated CO2 in the atmosphere, we treat their effects uniformly. Implications of this are discussed below.

SO2 present at soil -99% of the gas by volume; see, for example, data in (Sorey et al 1998, Werner et al, 2014). and a number of papers on volcanic decassing at Mammoth Mountain (Lewicki 2006 2008, 2012, 2014). Nitrogen is measured in small quantities (-1%).

Although the AVIRIS, MASTER, and ASO lidarairborne datasets cover a wider region, only points with associated soil CO2 flux measurements were used to derive our models. The CO2 flux measurements were spatially resampled to match the resolution of the other datasets. which resulted in small estimations with low confidence along the edges. To avoid spurious

model fits, edge points with  $CO_2 < 5 \text{ gr} \text{m}^{-2} \text{-} d^{-1}$  were excluded, where the  $CO_2$  range is [0,2000] g-m-2-d-1. In the remainder of this manuscript, analysed points with elevated CO2 flux will be referred to as eCO2.

**2.1.2 AVIRIS**

5

10

35

- The Airborne Visible/Infrared Imaging Spectrometer (AVIRIS) Classic instrument acquires 15 data from 400 to 2500 nm in 224 contiguous spectral channels. AVIRIS imagery was acquired over Mammoth in October 2014; this flight was chosen from a number of possible surveys of the area to minimize snow cover, and also because of its temporal proximity to the eCO2 ground measurements. The standard level 2 (L2) atmospherically corrected reflectance
- data (Thompson et al., 2015) was used (available from https://aviris.jpl.nasa.gov/), and the 20 data had a spatial resolution of 13 m. This data was collected as part of the NASA HyspIRI Preparatory Airborne Campaign.

**Vegetation indices**

- 25 Vegetation indices are commonly used as an indicator of vegetation health and/or greenness. While many vegetation indices are related, they differ enough to be considered independent variables. E.g. some account for soil moisture, others weight plant greenness more heavily. This was an exploratory effort in investigating the effects of CO2 on any measure of plant function, composition, and structure, and so we attempted to cover all avenues of 30
  - investigation. The following indices were derived from the AVIRIS spectral data:
    - The Normalized Difference Vegetation Index (NDVI)
    - Simple Ratio Index
    - Enhanced Vegetation Index
    - Red Edge Normalized Difference Vegetation Index ٠
    - Modified Red Edge Simple Ratio Index
      - Modified Red Edge Normalized Difference Vegetation Index

[revised manuscript text omitted]

- 20 Gerlach, T. M. (1991). Present day CO2 emissions from volcanos. *Eos, Transactions American Geophysical Union*, 72(23), 249–255. doi: 10.1029/90EO10192 Giammanco, S., Sims, K. W. W., & Neri, M. (2007). Measurements of 220Rn and 222Rn and
- CO2-emissions in soil and fumarole gases on Mt. Etna volcano (Italy): Implications for gas transport and shallow ground fracture. *Geochemistry, Geophysics, Geosystems,* 8(10) https://doi.org/10.1029/2007GC001644
- Gillespie, A., Rokugawa, S., Matsunaga, T., Cothern, J.S., Hook, S.J., and Kahle, A.B, 1998.
   A temperature and emissivity separation algorithm for Advanced Spaceborne Thermal Emission and Relection Radiometer (ASTER) images. *IEEE Transactions on Geoscience and Remote Sensing*, vol. 36, pp. 1113–1126.
- 30 Hernández, P. A., Pérez, N. M., Salazar, J. M., Nakai, S., Notsu, K., & Wakita, H. (1998). Diffuse emission of carbon dioxide, methane, and helium-3 from Teide Volcano, Tenerife, Canary Islands. *Geophysical Research Letters*, 25(17), 3311–3314. https://doi.org/10.1029/98GL02561
- Huang, J. G., Y. Bergeron, B. Denneler, F. Berninger, and J. Tardif (2007), Response of forest trees to increased atmospheric CO2, *Crit. Rev. Plant Sci.*, 26, 265–283, doi:10.1080/07352680701626978.

| Kerrick, D. M. (2001). Present and past nonanthropogenic CO2 degassing from the solid         |
|-----------------------------------------------------------------------------------------------|
| earth. Reviews of Geophysics, 39(4), 565–585.                                                 |
| https://doi.org/10.1029/2001RG000105                                                          |
| Kimball, B.A, LaMorte, R.L., Seay, R.S., Pinter, P.J. Jr, Rokey, R.R., Hunsaker, D.J., Dugas, |
|                                                                                               |

5 W.A., Heuer, M.L., Mauney, J.R., Hendrey, G.R., Lewin, K.F., Nagy, J. (1998). Effects of free air CO2 enrichment on energy balance and evapotranspiration of cotton. Agricultural and Forest Meteorology, 70, 259–278.

Kobayashi, H., and H. Iwabuchi (2008), A coupled 1-D atmosphere and 3-D canopy radiative transfer model for canopy reflectance, light environment, and photosynthesis

- 10 simulation in a heterogeneous landscape, *Remote Sensing of Environment*, 112(1), 173–185.
  - Kolby Smith, W., Reed, S. C., Cleveland, C. C., Ballantyne, A. P., Anderegg, W. R. L., Wieder, W. R., ... Running, S. W. (2015). Large divergence of satellite and Earth system model estimates of global terrestrial CO2 fertilization. *Nature Climate*
- 15 *Change*, 6, 306. Retrieved from http://dx.doi.org/10.1038/nclimate2879 Körner, C. (2006). Plant CO2 responses: an issue of definition, time and resource supply. *New Phytologist*, 172(3), 393–411. doi: 10.1111/j.1469-8137.2006.01886.x

20

Le Quéré, C., Andrew, R. M., Canadell, J. G., Sitch, S., Korsbakken, J. I., Peters, G. P., ... Zachle, S. (2016). Global Carbon Budget 2016. *Earth Syst. Sci. Data*, 8(2), 605–649. doi: 10.5194/essd 8-605-2016

- Lewicki, J. L., M.L. Fischer, and G.E. Hilley, 2008. Six-week time series of eddy covariance CO2 flux at Mammoth Mountain, California: performance evaluation and role of meteorological forcing, *Journal of Volcanology and Geothermal Research*, 171, 178-190, doi:10.1016/j.jvolgeores.2007.11.029.
- 25 Lewicki, J.L. and G.E. Hilley (2014a). Multi-scale observations of the variability of magmatic CO2 emissions, Mammoth Mountain, CA, USA. *Journal of Volcanology* and Geothermal Research, 284, 1–15, doi:10.1016/j.jvolgeores.2014.07.011.
  - Lewicki, J.L., Hilley, G.E., Shelly, D.R., King, J.C., McGeehin, J.P., Mangan, M. (2014b). Crustal migration of CO2-rich magmatic fluids recorded by tree-ring radiocarbon and
- 30 seismicity at Mammoth Mountain, CA, USA. *Earth and Planetary Science Letters*, 390, 52-58.
  - Lewin, K. F., Hendrey, G. R., Nagy, J., & LaMorte, R. L. (1994). Design and application of a free-air carbon dioxide enrichment facility. *Agricultural and Forest Meteorology*, 70(1), 15–29. https://doi.org/https://doi.org/10.1016/0168-1923(94)90045-0
- 35 List, M.R. (2005). Geologic remote sensing and GIS evaluation of volcanogenic CO2-induced tree kills: Mammoth Mountain, California, 1991–2004, M.S. Thesis, CSUS.

- Ma, P., Kang, E. L., Braverman, A., & Nguyen, H. (2018). Spatial statistical downscaling for constructing high resolution nature runs in global observing system simulation experiments. In review.
- Mason, E., Edmonds, M. and Turchyn, A.V., 2017. Remobilization of crustal carbon may dominate volcanic arc emissions. Science, 357(6348), pp.290–294.
- McGee, K. A., & Gerlach, T. M. (1998). Annual cycle of magmatic CO2 in a tree kill soil at Mammoth Mountain, California: Implications for soil acidification. *Geology*, 26(5), 463–466. Retrieved from http://dx.doi.org/10.1130/0091-7613(1998)026%3C0463:ACOMCI%3E2.3.CO
- 10 McGuire, A. D., Melillo, J. M., & Joyce, L. A. (1995). The Role of Nitrogen in the Response of Forest Net Primary Production to Elevated Atmospheric Carbon Dioxide. Annual Review of Ecology and Systematics, 26, 473–503. Retrieved from http://harvardforest.fas.harvard.edu/sites/harvardforest.fas.harvard.edu/files/publicatio ns/pdfs/McGuire\_AnReviewEco&Sys\_1995.pdf
- 15 Nendel, C., Kersebaum, K.C., Mirschel, W., Manderscheid, R., Weigel, H. J., Wenkel, K. O. (2009). Testing different CO2-response algorithms against a FACE crop rotation experiment. NJAS — Wageningen Journal of Life Sciences, 57(1), 17–25.
- Norby, R. J., De Kauwe, M. G., Domingues, T. F., Duursma, R. A., Ellsworth, D. S., Goll, D. S., ... Zaehle, S. (2016). Model data synthesis for the next generation of forest free air CO2 enrichment (FACE) experiments. *New Phytologist*, 209(1), 17–28. doi: 10.1111/nph.13593
  - Ogretim, E., Crandall, D., Gray, D.D., Bromhai, G.S. 2013. Effects of Atmospheric Dynamics on CO2 Seepage at Mammoth Mountain, California USA. *Journal of Computational Multiphase Flows*, 5(4), 283-294.
- 25 Painter, T.H., Berisford, D.F., Boardman, J.W., Bormann, K.J., Deems, J.S., Gehrke, F., Hedrick, A., Joyce, M., Laidlaw, R., Marks, D., Mattmann, C., McGurk, B., Ramirez, P., Richardson, M., Skiles, S.M., Seidel, F.C., Winstral, A., 2016. The Airborne Snow Observatory: Fusion of scanning lidar, imaging spectrometer, and physically based modeling for mapping snow water equivalent and snow albedo. *Remote Sensing of*
- 30 *Environment* 184, 139–152. https://doi.org/10.1016/j.rse.2016.06.018 Perez, N. M., Hernandez, P. A., Padilla, G., Nolasco, D., Barrancos, J., Melian, G., ... Ibarra, M. (2011). Global CO2 emission from volcanic lakes. *Geology*, 39, 235–238.
- Peterson, B. J., & Melillo, J. M. (1985). The potential storage of carbon caused by eutrophication of the biosphere. *Tellus B*, *37B*(3), 117–127. doi: 10.1111/j.1600-35 0889.1985.tb00060.x
- Pickles, W.L., Kasameyer, P.W., Martini, B.A., Potts, D.C., Silver, E.A. 2001. Geobotanical Remote Sensing for Geothermal Exploration. Geothermal Resources Council 2001 Annual Meeting, San Diego, California, August 26-29, 2001.

[revised manuscript text omitted]

Field Code Changed

|                      | Drake, B. G., Gonzàlez-Meler, M. A., and Long, S. P.: More Efficient Plants: A Consequence                                                                                                                                                                                                                                                                                                                                                                                                                                                                                                                                                                                                                                                                                                                                                                                                                                                                                                                                                                                                                                                                                                                                                                                                                                      |                    |  |
|----------------------|---------------------------------------------------------------------------------------------------------------------------------------------------------------------------------------------------------------------------------------------------------------------------------------------------------------------------------------------------------------------------------------------------------------------------------------------------------------------------------------------------------------------------------------------------------------------------------------------------------------------------------------------------------------------------------------------------------------------------------------------------------------------------------------------------------------------------------------------------------------------------------------------------------------------------------------------------------------------------------------------------------------------------------------------------------------------------------------------------------------------------------------------------------------------------------------------------------------------------------------------------------------------------------------------------------------------------------|--------------------|--|
|                      | of Rising Atmospheric CO2? Annu. Rev. Plant Phys., 48(1), 609–639. doi:                                                                                                                                                                                                                                                                                                                                                                                                                                                                                                                                                                                                                                                                                                                                                                                                                                                                                                                                                                                                                                                                                                                                                                                                                                                         |                    |  |
|                      | 10.1146/annurev.arplant.48.1.609, 1997.                                                                                                                                                                                                                                                                                                                                                                                                                                                                                                                                                                                                                                                                                                                                                                                                                                                                                                                                                                                                                                                                                                                                                                                                                                                                                         |                    |  |
|                      | Evans, J. S., Hudak, A. T.: A multiscale curvature algorithm for classifying discrete return                                                                                                                                                                                                                                                                                                                                                                                                                                                                                                                                                                                                                                                                                                                                                                                                                                                                                                                                                                                                                                                                                                                                                                                                                                    |                    |  |
| 5                    | LiDAR in forested environments. IEEE T. Geosci. Remote, 45, 1029–1038.                                                                                                                                                                                                                                                                                                                                                                                                                                                                                                                                                                                                                                                                                                                                                                                                                                                                                                                                                                                                                                                                                                                                                                                                                                                          |                    |  |
|                      | doi:http://dx.doi.org/10.1109/TGRS.2006.890412, 2007.                                                                                                                                                                                                                                                                                                                                                                                                                                                                                                                                                                                                                                                                                                                                                                                                                                                                                                                                                                                                                                                                                                                                                                                                                                                                           |                    |  |
|                      | Farrar, C. D., Sorey, M. L., Evans, W. C., Howle, J. F., Kerr, B. D., Kennedy, B. M.,                                                                                                                                                                                                                                                                                                                                                                                                                                                                                                                                                                                                                                                                                                                                                                                                                                                                                                                                                                                                                                                                                                                                                                                                                                           |                    |  |
|                      | Southon, J. R.: Forest-killing diffuse CO2 emission at Mammoth Mountain as a sign                                                                                                                                                                                                                                                                                                                                                                                                                                                                                                                                                                                                                                                                                                                                                                                                                                                                                                                                                                                                                                                                                                                                                                                                                                               |                    |  |
|                      | of magmatic unrest. Nature, 376(6542), 675–678. Retrieved from                                                                                                                                                                                                                                                                                                                                                                                                                                                                                                                                                                                                                                                                                                                                                                                                                                                                                                                                                                                                                                                                                                                                                                                                                                                                  |                    |  |
| 10                   | http://dx.doi.org/10.1038/376675a0, 1995.                                                                                                                                                                                                                                                                                                                                                                                                                                                                                                                                                                                                                                                                                                                                                                                                                                                                                                                                                                                                                                                                                                                                                                                                                                                                                       |                    |  |
|                      | Fisher, J. B., and ECOSTRESS Algorithm Development Team.: ECOsystem Spaceborne                                                                                                                                                                                                                                                                                                                                                                                                                                                                                                                                                                                                                                                                                                                                                                                                                                                                                                                                                                                                                                                                                                                                                                                                                                                  | Field Code Changed |  |
|                      | Thermal Radiometer Experiment on Space Station (ECOSTRESS): Level-3                                                                                                                                                                                                                                                                                                                                                                                                                                                                                                                                                                                                                                                                                                                                                                                                                                                                                                                                                                                                                                                                                                                                                                                                                                                             |                    |  |
|                      | Evapotranspiration Algorithm Theoretical Basis Document, 24 pp, Jet Propulsion                                                                                                                                                                                                                                                                                                                                                                                                                                                                                                                                                                                                                                                                                                                                                                                                                                                                                                                                                                                                                                                                                                                                                                                                                                                  |                    |  |
|                      | Laboratory, Pasadena, 2015.                                                                                                                                                                                                                                                                                                                                                                                                                                                                                                                                                                                                                                                                                                                                                                                                                                                                                                                                                                                                                                                                                                                                                                                                                                                                                                     |                    |  |
| 15                   | Fisher, J. B., Melton, F., Middleton, E., Hain, C., Anderson, M., Allen, R., McCabe, M. F.,                                                                                                                                                                                                                                                                                                                                                                                                                                                                                                                                                                                                                                                                                                                                                                                                                                                                                                                                                                                                                                                                                                                                                                                                                                     |                    |  |
|                      | Hook, Baldocchi, D., Townsend, P. A., Kilic, A., Tu, K., Miralles, D. D., Perret, J.,                                                                                                                                                                                                                                                                                                                                                                                                                                                                                                                                                                                                                                                                                                                                                                                                                                                                                                                                                                                                                                                                                                                                                                                                                                           |                    |  |
|                      | Lagouarde, JP., Waliser, D., Purdy, A. J., French, A., Schimel, D., Famiglietti, J. S.,                                                                                                                                                                                                                                                                                                                                                                                                                                                                                                                                                                                                                                                                                                                                                                                                                                                                                                                                                                                                                                                                                                                                                                                                                                         |                    |  |
|                      | Stephens, G., Wood, E. F.: The Future of Evapotranspiration: Global requirements for                                                                                                                                                                                                                                                                                                                                                                                                                                                                                                                                                                                                                                                                                                                                                                                                                                                                                                                                                                                                                                                                                                                                                                                                                                            |                    |  |
|                      | ecosystem functioning, carbon and climate feedbacks, agricultural management, and                                                                                                                                                                                                                                                                                                                                                                                                                                                                                                                                                                                                                                                                                                                                                                                                                                                                                                                                                                                                                                                                                                                                                                                                                                               |                    |  |
| • •                  |                                                                                                                                                                                                                                                                                                                                                                                                                                                                                                                                                                                                                                                                                                                                                                                                                                                                                                                                                                                                                                                                                                                                                                                                                                                                                                                                 |                    |  |
| 20                   | water resources. Water Resour. Res., 53(4), 2618-2626, 2017.                                                                                                                                                                                                                                                                                                                                                                                                                                                                                                                                                                                                                                                                                                                                                                                                                                                                                                                                                                                                                                                                                                                                                                                                                                                                    |                    |  |
| 20                   | water resources. Water Resour. Res., 53(4), 2618-2626, 2017.
Fisher, J. B., Tu, K., Baldocchi, D. D.: Global estimates of the land-atmosphere water flux                                                                                                                                                                                                                                                                                                                                                                                                                                                                                                                                                                                                                                                                                                                                                                                                                                                                                                                                                                                                                                                                                                                                                                     | Field Code Changed |  |
| 20                   | water resources. Water Resour. Res., 53(4), 2618-2626, 2017.
Fisher, J. B., Tu, K., Baldocchi, D. D.: Global estimates of the land-atmosphere water flux
based on monthly AVHRR and ISLSCP-II data, validated at 16 FLUXNET
sites. Remote Sana Environ. 112, 201, 010, 2008                                                                                                                                                                                                                                                                                                                                                                                                                                                                                                                                                                                                                                                                                                                                                                                                                                                                                                                                                                                                                                            | Field Code Changed |  |
| 20                   | water resources. Water Resour. Res., 53(4), 2618-2626, 2017.
Fisher, J. B., Tu, K., Baldocchi, D. D.: Global estimates of the land-atmosphere water flux
based on monthly AVHRR and ISLSCP-II data, validated at 16 FLUXNET
sites. Remote Sens. Environ., 112, 901-919, 2008.
Friedlingstein P. Meinshussen M. Argen V. K. Janes, C. D. Angu, A. Liddigest, S. K.                                                                                                                                                                                                                                                                                                                                                                                                                                                                                                                                                                                                                                                                                                                                                                                                                                                                                                                                                   | Field Code Changed |  |
| 20                   |  <li>water resources. Water Resour. Res., 53(4), 2618-2626, 2017.</li> <li>Fisher, J. B., Tu, K., Baldocchi, D. D.: Global estimates of the land-atmosphere water flux
based on monthly AVHRR and ISLSCP-II data, validated at 16 FLUXNET
sites. Remote Sens. Environ., 112, 901-919, 2008.</li> <li>Friedlingstein, P., Meinshausen, M., Arora, V. K., Jones, C. D., Anav, A., Liddicoat, S. K.,
Knutti, P.: Uncertainties in CMUP5 Climate Projections due to Carbon Cycle</li>                                                                                                                                                                                                                                                                                                                                                                                                                                                                                                                                                                                                                                                                                                                                                                                                                          | Field Code Changed |  |
| 20
25             |  <li>water resources. Water Resour. Res., 53(4), 2618-2626, 2017.</li> <li>Fisher, J. B., Tu, K., Baldocchi, D. D.: Global estimates of the land-atmosphere water flux
based on monthly AVHRR and ISLSCP-II data, validated at 16 FLUXNET
sites. Remote Sens. Environ., 112, 901-919, 2008.</li> <li>Friedlingstein, P., Meinshausen, M., Arora, V. K., Jones, C. D., Anav, A., Liddicoat, S. K.,
Knutti, R.: Uncertainties in CMIP5 Climate Projections due to Carbon Cycle
Eeedbacks. J. Climate. 27, 512-526, 2014.</li>                                                                                                                                                                                                                                                                                                                                                                                                                                                                                                                                                                                                                                                                                                                                                                            | Field Code Changed |  |
| 20
25             |  <li>water resources. Water Resour. Res., 53(4), 2618-2626, 2017.</li> <li>Fisher, J. B., Tu, K., Baldocchi, D. D.: Global estimates of the land-atmosphere water flux
based on monthly AVHRR and ISLSCP-II data, validated at 16 FLUXNET
sites. Remote Sens. Environ., 112, 901-919, 2008.</li> <li>Friedlingstein, P., Meinshausen, M., Arora, V. K., Jones, C. D., Anav, A., Liddicoat, S. K.,
Knutti, R.: Uncertainties in CMIP5 Climate Projections due to Carbon Cycle
Feedbacks. J. Climate, 27, 512-526, 2014.</li> <li>Garcia M. Saatchi, S. Ferraz, A. Silva, C.A. Ulstin, S. Koltunov, A. Balzter, H.: Impact of</li>                                                                                                                                                                                                                                                                                                                                                                                                                                                                                                                                                                                                                                                                       | Field Code Changed |  |
| 20
25             |  <li>water resources. Water Resour. Res., 53(4), 2618-2626, 2017.</li> <li>Fisher, J. B., Tu, K., Baldocchi, D. D.: Global estimates of the land-atmosphere water flux
based on monthly AVHRR and ISLSCP-II data, validated at 16 FLUXNET
sites. Remote Sens. Environ., 112, 901-919, 2008.</li> <li>Friedlingstein, P., Meinshausen, M., Arora, V. K., Jones, C. D., Anav, A., Liddicoat, S. K.,
Knutti, R.: Uncertainties in CMIP5 Climate Projections due to Carbon Cycle
Feedbacks. J. Climate, 27, 512-526, 2014.</li> <li>Garcia, M., Saatchi, S., Ferraz, A., Silva, C.A., Ustin, S., Koltunov, A., Balzter, H.: Impact of
data model and point density on aboveground forest biomass estimation from airborne.</li>                                                                                                                                                                                                                                                                                                                                                                                                                                                                                                                                                                        | Field Code Changed |  |
| 20
25             |  <li>water resources. Water Resour. Res., 53(4), 2618-2626, 2017.</li> <li>Fisher, J. B., Tu, K., Baldocchi, D. D.: Global estimates of the land-atmosphere water flux
based on monthly AVHRR and ISLSCP-II data, validated at 16 FLUXNET
sites. Remote Sens. Environ., 112, 901-919, 2008.</li> <li>Friedlingstein, P., Meinshausen, M., Arora, V. K., Jones, C. D., Anav, A., Liddicoat, S. K.,
Knutti, R.: Uncertainties in CMIP5 Climate Projections due to Carbon Cycle
Feedbacks. J. Climate, 27, 512-526, 2014.</li> <li>Garcia, M., Saatchi, S., Ferraz, A., Silva, C.A., Ustin, S., Koltunov, A., Balzter, H.: Impact of
data model and point density on aboveground forest biomass estimation from airborne
LiDAR. Carbon Balance and Management, 12, doi:10.1186/s13021-017-0073-1.</li>                                                                                                                                                                                                                                                                                                                                                                                                                                                                                            | Field Code Changed |  |
| 20
25
30       |  <li>water resources. Water Resour. Res., 53(4), 2618-2626, 2017.</li> <li>Fisher, J. B., Tu, K., Baldocchi, D. D.: Global estimates of the land-atmosphere water flux
based on monthly AVHRR and ISLSCP-II data, validated at 16 FLUXNET
sites. Remote Sens. Environ., 112, 901-919, 2008.</li> <li>Friedlingstein, P., Meinshausen, M., Arora, V. K., Jones, C. D., Anav, A., Liddicoat, S. K.,
Knutti, R.: Uncertainties in CMIP5 Climate Projections due to Carbon Cycle
Feedbacks. J. Climate, 27, 512-526, 2014.</li> <li>Garcia, M., Saatchi, S., Ferraz, A., Silva, C.A., Ustin, S., Koltunov, A., Balzter, H.: Impact of
data model and point density on aboveground forest biomass estimation from airborne
LiDAR. Carbon Balance and Management, 12. doi:10.1186/s13021-017-0073-1,
2017.</li>                                                                                                                                                                                                                                                                                                                                                                                                                                                                                  | Field Code Changed |  |
| 20
25
30       |  <li>water resources. Water Resour. Res., 53(4), 2618-2626, 2017.</li> <li>Fisher, J. B., Tu, K., Baldocchi, D. D.: Global estimates of the land-atmosphere water flux
based on monthly AVHRR and ISLSCP-II data, validated at 16 FLUXNET
sites. Remote Sens. Environ., 112, 901-919, 2008.</li> <li>Friedlingstein, P., Meinshausen, M., Arora, V. K., Jones, C. D., Anav, A., Liddicoat, S. K.,
Knutti, R.: Uncertainties in CMIP5 Climate Projections due to Carbon Cycle
Feedbacks. J. Climate, 27, 512-526, 2014.</li> <li>Garcia, M., Saatchi, S., Ferraz, A., Silva, C.A., Ustin, S., Koltunov, A., Balzter, H.: Impact of
data model and point density on aboveground forest biomass estimation from airborne
LiDAR. Carbon Balance and Management, 12. doi:10.1186/s13021-017-0073-1,
2017.</li> <li>Gerlach, T. M.: Present-day CO2 emissions from volcanos. Eos Trans AGU, 72(23), 249–</li>                                                                                                                                                                                                                                                                                                                                                                         | Field Code Changed |  |
| 20
25
30       |  <li>water resources. Water Resour. Res., 53(4), 2618-2626, 2017.</li> <li>
[revised manuscript text omitted]

---

## Author Response (AR3)

Dear Prof. Schöngart,

Thank you for your technical corrections, and your continued investment in this manuscript. We have made all changes suggested, as outlined below.

Technical corrections of bg-2018-73:

In terms of in-text citations, the order can be based on relevance, as well as chronological or alphabetical listing, depending on the author's preference, however, it
10 should me consistent. In some parts in-text citations are listed from the most recent study towards the most ancient, in other place it is vice versa.
This has been corrected. References are now ordered in decreasing chronological order consistently.

15 There is no consistent use of capitals for words in the titles of sections.
This has been corrected. Title case has been used consistently.

Material and methods:
P. 6/L. 5-6: Is "Classic instrument" correct or necessary in this sentence?
20 Yes. This is to differentiate between the instruments "AVIRIS Classic" and "AVIRIS Next Generation".

P. 6/L. 17: "E.g." at the beginning of the sentence should be avoided
This sentence has been rephrased.
25
P.12/ L. 4: add a comma after et al.
Done.

P. 12/L/.18: What does "<<0.05" means? Below 0.01?
30 Yes, some are on the order of 1e-8, some even lower (there is a wide range). I have changed the text to <<0.01.

Discussion:
P. 15/L. 24: Please delete the comma after et al.
35 Done.

P. 17/L. 23: Should be "Tissue et al., 1999)."
This has been corrected.

In Figure 1, I am wondering about the atmospheric CO2 flux (blue line). Why is it constant over the 25-yr period?

The line represents a single data point measured in 2014, and was marked as a line for clarity in the figure (as noted in the caption). However, since that seems to cause confusion, I have reverted back to a single point.

[revised manuscript text omitted]